# Eliminating separase inhibition reveals absence of robust cohesin protection in oocyte metaphase II

Safia El Jailani [ID][1], Damien Cladière [ID][1], Elvira Nikalayevich[2,4], Sandra A Touati [ID][1], Vera Chesnokova[3], Shlomo Melmed[3], Eulalie Buffin[1,2] & Katja Wassmann [ID][1✉]

## Abstract

The meiotic segregation pattern to generate haploid gametes is mediated by step-wise cohesion removal by separase, first from chromosome arms in meiosis I, and then from the pericentromere in meiosis II. In mammalian oocytes, separase is tightly controlled during the hours-long prometaphase and until chromosome segregation in meiosis I, activated for a short time window, and again inhibited until metaphase II arrest is lifted by fertilization. Centromeric cohesin is protected from cleavage by Sgo2-PP2A in meiosis I. It remained enigmatic how tight control of alternating separase activation and inactivation is achieved during the two divisions in oocytes, and when cohesin protection is put in place and removed. Using complementation assays in knock-out mouse models, we established the contributions of cyclin B1 and securin for separase inhibition during both divisions. When eliminating separase inhibition, we found that cohesin is not robustly protected at meiosis I resumption and during metaphase II arrest. Importantly, in meiosis II, the sole event required for cleavage of peri-centromeric cohesin besides separase activation is prior kinetochore individualization in meiosis I.

**Keywords** Cohesin Protection; Kinetochore Individualization; Meiosis; Oocytes; Separase Inhibition
**Subject Categories** Cell Cycle; Chromatin, Transcription & Genomics; Post-translational Modifications & Proteolysis

## Introduction

Correct chromosome segregation during meiosis is essential to generate gametes with the correct ploidy and thus, healthy offspring. Unlike mitosis, during which sister chromatids are segregated to opposite poles to generate two identical daughter cells, meiosis requires segregation of homologous chromosomes in meiosis I and of sister chromatids in meiosis II. Unless pericentromeric cohesin is maintained until meiosis II, sister chromatids separate randomly in anaphase II. This is due to the failed establishment of bipolar, tension-bearing attachments when there is no physical connection between the two sister chromatids. The protease separase is key to remove cohesion holding sister chromatids together. Separase acts by cleaving the kleisin subunit of the cohesin complex, namely Scc1 in mitosis and Rec8 in both meiotic divisions. In meiosis, cohesin cleavage takes place in a step-wise manner: at chromosome arms to resolve chiasmata (sites of recombination) in meiosis I, and at the pericentromere in meiosis II. Thus, centromeric cohesin requires to be protected from separase cleavage in meiosis I, and deprotected for proper execution of meiosis II (Marston and Amon, 2004; Petronczki et al, 2003).

Cohesin cleavage in meiosis is controlled by phosphorylation of Rec8, a pre-requisite for efficient separase cleavage, and by inhibitors directly impinging on separase and keeping separase inactive until satisfaction of the spindle assembly checkpoint (SAC) and anaphase onset (Konecna et al, 2023; Wassmann, 2022). Rec8 phosphorylation at the pericentromere in meiosis I is prevented by localization of the phosphatase PP2A-B56 through Sgo2 recruitment, thus protecting Rec8 from cleavage during the first meiotic division (Gutierrez-Caballero et al, 2012; Marston, 2015). We recently showed that a physical separation of sister kinetochores (sister kinetochore individualization) prior to the meiosis I-to-meiosis II transition is key for deprotection and cleavage of pericentromeric cohesin in meiosis II (Gryaznova et al, 2021). In an independent study, a third fraction of Rec8—localized at the centromere—was shown to be cleaved once arm cohesin has been removed in anaphase I, and to be required for pericentromeric cohesin cleavage in meiosis II (Ogushi et al, 2021). In budding yeast, pericentromeric cohesin protection is eliminated at the metaphase-to-anaphase transition of meiosis II in an APC/C (Anaphase Promoting Complex/ Cyclosome)-dependent manner (Jonak et al, 2017; Mengoli et al, 2021). It is yet unclear whether or not in oocytes, which have to maintain an extended arrest in metaphase II to await fertilization, cohesin deprotection is needed and equally requires APC/C activity.

In terms of the cell cycle, the meiotic divisions constitute a real challenge, as cells have to execute two M-phases without interphase and the associated reset of the cell cycle in between. In addition, oocytes await fertilization during a prolonged metaphase II arrest of

[1]Université Paris Cité, CNRS, Institut Jacques Monod, 75013 Paris, France. [2]IBPS, Sorbonne Université; CNRS UMR 7622, Sorbonne Université, 75252 Paris, France. [3]Pituitary Center, Department of Medicine, Cedars-Sinai Medical Center, Los Angeles, CA 90048, USA. [4]Present address: CIRB, Collège de France, UMR7241/U1050, 75005 Paris, France. ✉E-mail: katja.wassmann@ijm.fr

variable length, before lift of the arrest and anaphase II onset only upon sperm entry (Holt et al, 2013). Oocytes are not renewed from birth of the female until fertilization, placing incredible temporal strain on the development of female gametes. This leads to the deterioration of cohesin complexes and SAC control, and the generation of aneuploidies with increasing maternal age (Herbert et al, 2015). It is therefore important to understand how separase activity, cell cycle progression and step-wise cohesin removal are coordinated in mammalian oocyte meiosis, to understand how errors arise.

Separase has to be tightly controlled until anaphase I onset, transiently activated to allow removal of arm cohesin, and re-inhibited for entry into meiosis II. Again, at entry into meiosis II, re-inhibition of separase must be very tight but at the same time readily reversible when fertilization occurs (Konecna et al, 2023; Wassmann, 2022). In prophase of mitosis, separase is excluded from the nucleus due to a nuclear export signal (Hellmuth et al, 2018; Sun et al, 2006). Thus, separase inhibition becomes important once nuclear envelope breakdown occurs. Importantly, in oocytes, separase is not physically separated from its substrate at the transition from meiosis I into meiosis II, because no nucleus is reformed between the two meiotic divisions. It is currently unclear how separase inhibition is guaranteed at this time.

Cyclin B1 and securin, which both inhibit vertebrate separase in mitosis and meiosis, are ubiquitinated by the APC/C and thus targeted for degradation prior to anaphase onset (Kamenz and Hauf, 2017; Yu et al, 2023). Of note, securin, which was discovered as pituitary tumor transforming gene (PTTG) (Pei and Melmed, 1997), is both a separase inhibitor and a chaperone for separase, and therefore contributes to full separase activity (Holland and Taylor, 2008). For separase inhibition, securin acts as a pseudo-substrate and blocks separase by binding to its substrate pocket (Lin et al, 2016; Waizenegger et al, 2002; Yu et al, 2021). Cyclin B1, together with Cdk1, inhibits separase through the phosphorylation of a conserved residue, creating a pocket for the inhibitory binding of cyclin B1-Cdk1 to separase (Gorr et al, 2005; Holland and Taylor, 2006; Stemmann et al, 2001; Yu et al, 2021). Separase inhibition by securin or cyclin B1 is mutually exclusive. Initially, it was proposed that cyclin B1-dependent inhibition of separase occurs concomitantly with securin (Stemmann et al, 2001). In addition, cyclin B1 inhibition of separase may become important once securin is degraded, because complete cyclin B1 degradation is temporally delayed compared to securin (Afonso et al, 2019; Collin et al, 2013; Shindo et al, 2012; Wolf et al, 2006; Yu et al, 2023). Cyclin B1-Cdk1 activity itself is also inhibited when bound to separase, which may thus contribute to down-regulation of Cdk1 activity at exit from mitosis (Gorr et al, 2006) and potentially, at the transition from meiosis I into meiosis II. In the absence of securin, cyclin B1-Cdk1 seems to efficiently phosphorylate and inhibit securin-free separase in mitotic prometaphase, explaining why loss of securin -either in tissue culture cells or in a mouse model- is not lethal (Jallepalli et al, 2001; Mei et al, 2001; Pfleghaar et al, 2005; Wang et al, 2003; Wang et al, 2001).

Both inhibitors control separase activity in oocyte meiosis, and degradation of both is required for metaphase-to-anaphase transition in meiosis I (Herbert et al, 2003; Terret et al, 2003). Securin was proposed to function as the main separase inhibitor in oocyte meiosis II (Nabti et al, 2008), at odds with the fact that mice null for securin are viable and fertile (Mei et al, 2001; Wang et al,

2001). Together, the relative contributions and potential redundancies of separase inhibition by either securin or cyclin B1 in meiosis I and meiosis II are still unknown.

A third inhibitor of separase, namely Sgo2-Mad2, forms a complex with separase and is present in somatic cells arrested in mitosis due to SAC activation (Hellmuth et al, 2020). Whether this inhibitor has a physiological role in cycling cells or in meiosis to keep separase in check, is currently unknown. Sgo2 occupies multiple roles in oocyte meiosis, beyond centromeric cohesin protection. Complete loss of Sgo2 leads to delayed anaphase I onset, due to its function in silencing the spindle checkpoint. Segregation occurs only after APC/C activation in meiosis I, and there are no indications for precociously activated separase without Sgo2 (Marston, 2015; Rattani et al, 2013). Indeed, a recent study using transient approaches shows that Sgo2-Mad2 is not mediating separase inhibition in mouse oocyte meiosis I (Wetherall et al, 2025).

It is still unclear when pericentromeric cohesin protection is removed after anaphase I. Maintaining cohesin protected from cleavage during the transition from meiosis I into meiosis II may be important to prevent precocious sister chromatid segregation before reaccumulation of securin and cyclin B1 in meiosis II to inactivate separase. However, most of endogenous Sgo2 supposedly required for PP2A recruitment and cohesin protection is being displaced from pericentromeric chromatin at meiosis I exit (Gryaznova et al, 2021; Mengoli et al, 2021) before being recruited there again at high levels as oocytes progress into metaphase II (Chambon et al, 2013; Gryaznova et al, 2021; Lee et al, 2008; Mengoli et al, 2021; Mihalas et al, 2024). In the light of these results, pericentromeric cohesin protection was proposed to be in place in metaphase II (Chambon et al, 2013; Lee et al, 2008; Mengoli et al, 2021). Maintenance of pericentromeric cohesin at the meiosis I to meiosis II transition may depend on very low and thus undetectable levels of Sgo2, and/or alternative mechanisms such as the SUMO pathway (Ding et al, 2018).

Several models have been proposed for deprotection of centromeric cohesin: In mammalian meiosis, bipolar tension applied by the spindle on sister kinetochores in meiosis II was thought to lead to the deprotection of centromeric cohesin, e.g., at a time when separase inhibitor(s) have been able to re-accumulate (Gomez et al, 2007; Lee et al, 2008); however this "deprotection by tension model" was not confirmed (Gryaznova et al, 2021; Mengoli et al, 2021). In budding yeast, APC/C activation leading to the degradation of Sgo1 and Mps1 was shown to mediate deprotection, but whether this also applies to oocytes was unknown (Jonak et al, 2017; Mengoli et al, 2021). We previously reported a role for Set/I2PP2A in promoting cohesin removal in meiosis II (Chambon et al, 2013); yet our recent, unpublished results show that Set/I2PP2A does so independently of cohesin protection (Keating et al, in preparation). More recently, it was found that separase activity in anaphase I towards Rec8 at centromeres and concomitant sister kinetochore individualization are key for cleavage of pericentromeric cohesin in the subsequent meiosis II division (Gryaznova et al, 2021; Ogushi et al, 2021).

Here, we set out to determine the contributions of the two main separase inhibitors, cyclin B1-Cdk1 and securin, for maintaining separase control during the first and second meiotic division in mammalian oocytes, and to understand when exactly

pericentromeric cohesin can be cleaved. Separase inhibition in oocytes is dose-dependent (Chiang et al, 2011), and it is thus difficult to obtain conclusive results with transient knock-down approaches and exogenous protein expression alone. Using knock-out mouse strains and rescue experiments, we found that either cyclin B1-Cdk1 or securin are sufficient to maintain separase in check up until metaphase in both meiosis I and II. However, at the transition from meiosis I into meiosis II, cyclin B1-Cdk1 rather than securin is essential to re-inhibit separase. Strikingly, loss of both inhibitory mechanisms in either meiotic division leads to immediate separase activation and cleavage of cohesin. Importantly, without securin and cyclin B1-dependent inhibition of separate, we could determine when efficient pericentromeric cohesin protection is in place during the meiotic divisions in oocytes. Unexpectedly, we found that protection is not yet properly set up after nuclear envelope breakdown in early prometaphase I. In addition, maintenance of pericentromeric cohesin is sensitive to active separase being present during the meiosis I-to-meiosis II transition. Surprisingly, pericentromeric cohesin protection is largely absent during the metaphase II arrest, when oocytes await fertilization. Altogether, our results depict critical moments during meiotic cell cycle progression prone to missegregation events and the generation of aneuploidies in the absence of tight separase control in oocytes.

## Results

### Loss of cyclin B1-mediated separase inhibition does not disturb bivalent segregation

We have shown previously that exogenously expressed separase carrying a nonphosphorylatable amino-acid substitution of the conserved residue targeted by cyclin B1-Cdk1 (mouse separase S1121A) is not inhibited by cyclin B1 anymore during oocyte meiosis I (Touati et al, 2012). However, it was unknown whether cyclin B1-dependent inhibition of separase plays an important role in a physiological context (i.e., in the presence of securin) up to metaphase I and beyond.

To clarify this issue in an undisputable manner we performed complementation assays using separase knock-out mouse oocytes. Mature oocytes can be harvested before resumption of meiosis I, in prophase I (also called GV stage). They can be injected with mRNAs to express proteins of our choice, and induced to undergo meiosis I in a synchronized manner in culture. After GVBD (Germinal Vesicle Breakdown, corresponds to nuclear envelope breakdown) oocytes undergo prometaphase and reach metaphase around 6 h after GVBD. Metaphase-to-anaphase transition of meiosis I with polar body (PB) extrusion takes place around 8 h after GVBD. This is followed by exit from meiosis I and progression into meiosis II. Oocytes reach metaphase II ~12 h after GVBD and remain arrested for up to 12 h (mouse), until fertilization occurs. Fertilization induces anaphase II onset and exit from meiosis II (Fig. 1A).

A mouse strain allowing the conditional oocyte-specific invalidation of separase was obtained through the Cre-LoxP system under the control of the oocyte-specific *Zona pellucida 3* (*Zp3*) promoter. This strain has been previously characterized (Kudo et al, 2006). For simplicity, oocytes of *separase$^{LoxP/LoxP}$ Zp3 Cre$^+$* mice

are designated as *sep$^{-/-}$*, and controls (*separase$^{LoxP/LoxP}$*) as *sep$^{+/+}$*. Oocytes deleted for endogenous separase were thus used to address whether expression of the separase S1121A mutant caused inappropriate separase activation before metaphase I or failure to re-inhibit separase after anaphase I.

Control (*sep$^{+/+}$*) and *sep$^{-/-}$* oocytes were injected with mRNA coding either for wild-type separase or separase S1121A. To follow separase activity by live imaging, we additionally expressed a separase activity sensor (Nikalayevich et al, 2018), based on a similar sensor initially described in (Shindo et al, 2012). This cleavage biosensor contains a separase cleavage site and is artificially localized to chromosomes. Upon cleavage of the sensor, co-localization of the two fluorochromes of the sensor is lost, leading to a change of color that allows us to determine separase activation as well as anaphase onset timing, because the sensor allows us to follow chromosome movements as well (Movie EV1). Whereas no cleavage of the sensor was observed in oocytes devoid of functional separase, cleavage of the sensor took place with similar timing in *sep$^{-/-}$* oocytes expressing wild-type or S1121A mutant separase (Figs. 1B–D and EV1A). Interestingly, not only wild type, but also S1121A mutant separase rescued PB extrusion, indicating that the role of separase in promoting cytokinesis is not only independent of its protease activity (Kudo et al, 2006), but also of cyclin B1 binding (Fig. EV1B). Of note, a small but still significant delay in anaphase I onset was observed when separase was expressed in addition to endogenous separase in wild-type but not in *sep$^{-/-}$* oocytes (Fig. 1D). The reasons for this are unclear, but may be related to mutual inhibition of separase and cyclin B1-Cdk1 activity, reducing endogenous Cdk1 activity required for progression through meiosis I, when wild-type separase is expressed additionally to endogenous separase (Gorr et al, 2005; Gorr et al, 2006; Shindo et al, 2012). Crucially, chromosome spreads performed after the metaphase-to-anaphase transition of meiosis I confirmed that bivalent chromosomes had been segregated into dyads under the rescue conditions, no matter whether wild-type or S1121A separase was expressed (Fig. 1E,F). Precocious sister chromatid segregation was not observed. These results indicate that cyclin B1-dependent inhibition of separase alone is not essential to correctly inhibit separase, at least until chromosome segregation occurs in meiosis I.

### Cyclin B1-Cdk1 becomes essential to inhibit separase after anaphase I

We asked whether re-inhibition of separase after meiosis I, occurred correctly in *sep$^{-/-}$* oocytes expressing S1121A mutant separase. This is an important question, because it has been observed that Sgo2, which is required for cohesin protection, disappears from the centromere region in late anaphase I, before strongly reaccumulating there again in meiosis II (Gryaznova et al, 2021; Mengoli et al, 2021). Furthermore, sister kinetochore individualization and cleavage of Rec8 at centromeres, a prerequisite for pericentromeric cohesin cleavage in meiosis II, also occurs already in late anaphase I (Gryaznova et al, 2021; Ogushi et al, 2021). Therefore, re-inhibition of separase is most likely important to avoid cleavage of potentially unprotected cohesin and as a consequence, precocious sister chromatid segregation at the transition from meiosis I into meiosis II. To answer this question,

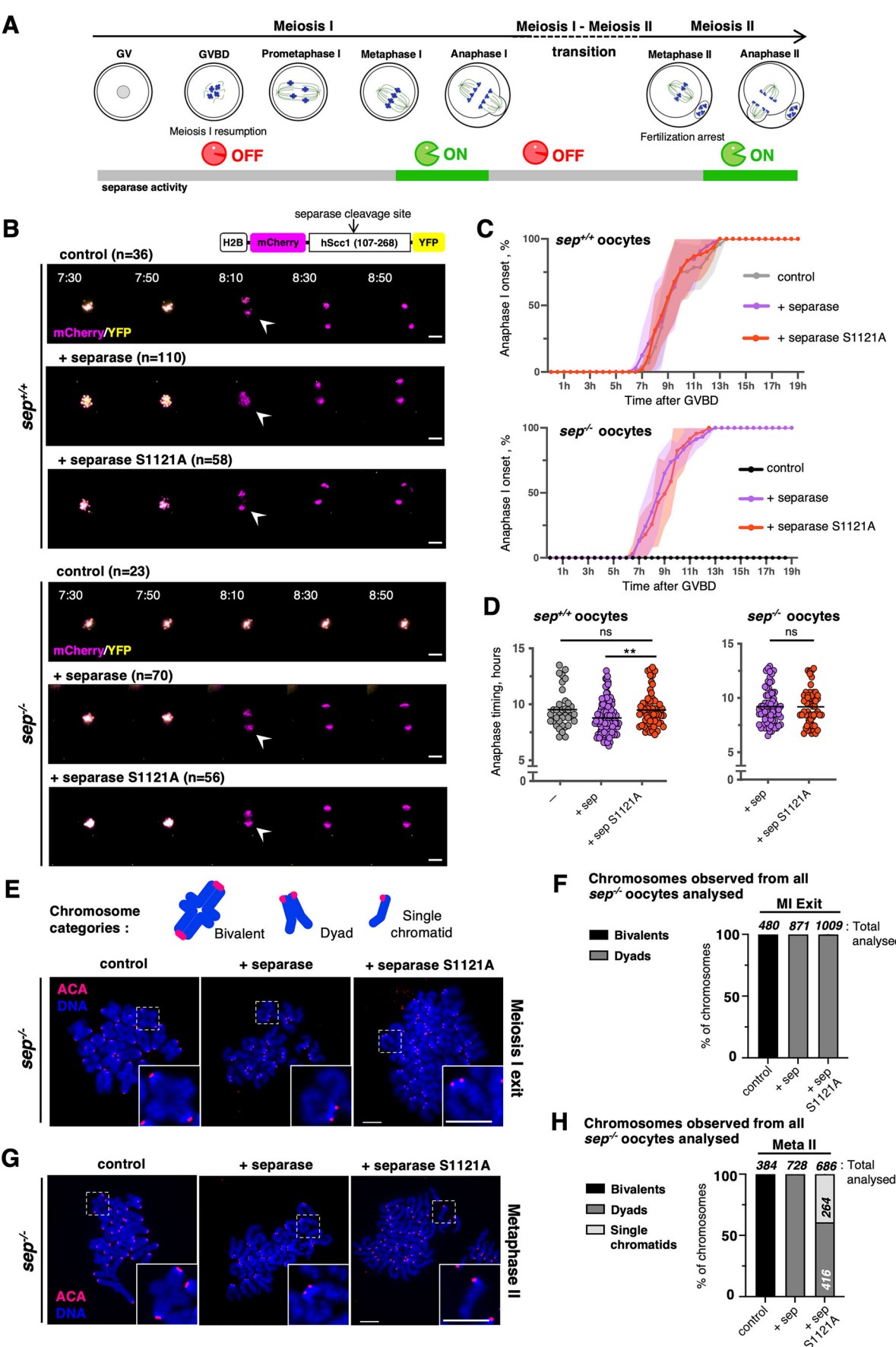

**Figure 1. Cyclin B1-dependent inhibition of separase becomes essential only after meiosis I exit.**

(A) Scheme of mouse oocyte meiosis I and II. Chromosomes are in blue, spindles in green, and the nucleus (Germinal Vesicle, GV) in gray. Separase activity at the different cell cycle stages is indicated below. (B) On top right, schematic design of the Scc1 cleavage sensor construct used. Below, overlays of the YFP and mCherry channels of time-lapse microscopy acquisitions of wild-type ($sep^{+/+}$) and separase knock-out ($sep^{-/-}$) mouse oocytes expressing the cleavage sensor. Time after GVBD is shown in hours:minutes (anaphase I onset is indicated with an arrowhead). Oocytes have been co-injected with mRNAs encoding for separase or separase S1121A, where indicated. Cleavage of the sensor is visible by the disappearance of the YFP signal from the chromosomes, whereas the mCherry signal remains localized to the chromosomes. $n$ is the number of oocytes analyzed. Scale bar (white) represents 20 μm (related to Fig. EV1A,B). (C) Monitoring the anaphase I onset of the selected time frames shown in (B). Hours after germinal vesicle breakdown (GVBD) are indicated. For each graph, error bars are ±SD. (D) Statistical analysis of anaphase timing of $sep^{+/+}$ and $sep^{-/-}$ oocytes from (B). For each graph, the mean is shown, ns indicates there is no significant difference, and asterisks indicate a significant difference (**$P = 0.0057$) according to Mann–Whitney $U$ test. (E) Top: representative schemes of chromosome categories observed, classified as "bivalent" (paired homologous chromosomes), "dyad" (paired sister chromatid), or "single chromatid". Bottom: $sep^{-/-}$ oocytes expressing separase or separase S1121A were fixed for chromosome spreads around 8 h after GVBD (Meiosis I exit). Oocytes were fixed after visual selection of oocytes extruding the polar body (PB), collection of $sep^{-/-}$ oocyte was time-matched. Centromeres/kinetochores were stained with ACA (red) and chromosomes with DAPI (blue). A representative spread and magnification of one chromosome (white dashed line squares, insert at the bottom right corner) are shown for each condition. The total number of chromosomes analyzed is indicated in (F). Scale bars (white) represent 10 μm. (F) The frequency of chromosome categories observed at exit from meiosis I (MI Exit), quantified from chromosome spreads shown in (E). Where indicated, $sep^{-/-}$ oocytes express separase or separase S1121A. The total number of chromosomes quantified for each condition is indicated. sep: separase. (G) $sep^{-/-}$ mouse oocytes expressing separase or separase S1121A were fixed for chromosome spreads around 20 h after GVBD (Metaphase II). $sep^{-/-}$ oocytes were collected at the same time. Centromeres/kinetochores were stained with ACA (red) and chromosomes with DAPI (blue). A representative spread and magnification of one chromosome (white dashed line squares, insert at the bottom right corner) are shown for each condition. The total number of chromosomes analyzed is indicated in (H). Scale bars (white) represent 10 μm. sep separase. (H) Frequency of chromosome categories observed at metaphase II arrest (MetaII), quantified from chromosome spreads in (G). Where indicated, $sep^{-/-}$ oocytes express separase or separase S1121A. The total number of chromosomes quantified for each condition, and of each category, is indicated. Data information: Results shown were obtained from at least three independent biological replicates. Source data are available online for this figure.

we repeated the previous rescue experiment, but performed chromosome spreads at a later timepoint, when oocytes had reached metaphase II and were awaiting fertilization. Crucially, now we observed a high fraction of single sister chromatids in $sep^{-/-}$ oocytes rescued with S1121A separase but not with wild-type separase, indicating that during the transition from meiosis I to meiosis II, separase remained active when cyclin B1-dependent inhibition was absent (Fig. 1G,H). These data also indicate that pericentromeric cohesin protection or SUMO-mediated maintenance of cohesin was not sufficient in the presence of constitutively active separase, leading to cleavage of cohesin at the pericentromere. We conclude that cyclin B1-dependent inhibition of separase is essential once anaphase I has taken place and oocytes progress into meiosis II.

## Securin is not essential in oocyte meiosis I

The nature of the inhibitory mechanism used by securin is not compatible with the generation of separase mutants that would only interfere with securin inhibition (Yu et al, 2023). For this reason, we decided to use oocytes derived from a complete securin knock-out ($Pttg^{-/-}$ mice (Wang et al, 2001), for clarity we call oocytes devoid of securin, $securin^{-/-}$ oocytes). Mice devoid of securin are viable, exhibit features of senescence, and demonstrate some signs of female subfertility (Chesnokova and Melmed, 2010; Mei et al, 2001; Wang et al, 2001). Oocytes from these complete knock-out mice resumed meiosis without delay. Separase activation and chromosome segregation occurred on time and without any noticeable issues, indicating that securin is not essential in meiosis I (Figs. 2A,B and EV2). A small delay in anaphase timing was observed in $securin^{-/-}$ oocytes; however, it was not statistically significant (Fig. 2C), indicating that the chaperone activity of securin (Holland and Taylor, 2008) is not important for full separase activity in oocyte meiosis. Hardly any precocious sister chromatid segregation was observed when $securin^{-/-}$ oocytes reached metaphase II arrest, indicating that securin is also not essential during the transition from meiosis I into meiosis II (Fig. 2D,E).

## "Separase-out-of-control" phenotype in prometaphase I

Are securin and cyclin B1 together the main separase inhibitors in meiosis? To clarify this issue, we created $separase^{LoxP/LoxP}$ $Zp3$ $Cre^+$ $Pttg^{-/-}$ mice. Oocytes are thus devoid of separase and securin ($sep^{-/-}$ $securin^{-/-}$ oocytes) and can be used for rescue experiments with separase S1121A. In these oocytes, separase is free of cyclin B1 and securin inhibition.

First, we asked how oocytes progress through meiosis I in the absence of both separase inhibitors. In $sep^{-/-}$ $securin^{-/-}$ oocytes rescued with wild-type separase, separase activation took place on time and anaphase I was visible. Strikingly, when separase S1121A was expressed, separase became activated immediately after resumption of meiosis I, shortly after germinal vesicle breakdown (GVBD), which corresponds to nuclear envelope breakdown (Fig. 3A). $sep^{-/-}$ $securin^{-/-}$ oocytes rescued with separase S1112A and followed by live imaging showed chaotic chromosome separation taking place, and cleavage of the separase sensor started shortly after GVBD, a phenotype we coined "separase-out-of-control" (Figs. 3A–C (compare also to Fig. 1B) and EV3A; Movie EV2, for comparison of stills of entire movies see Appendix Fig. S1). We conclude that separase must be inhibited by either cyclin B1 or securin in oocyte meiosis I, and that either inhibitor is sufficient, but when both are absent, separase immediately cleaves cohesin.

## Absence of efficient cohesin protection in early prometaphase I

Importantly, chromosome spreads in late prometaphase I revealed that the majority of bivalent chromosomes had segregated into sister chromatids when $sep^{-/-}$ $securin^{-/-}$ oocytes expressed separase S1121A. This was not the case when they expressed wild-type separase, which can still be inhibited by cyclin B1 (Fig. 3D,E). It was also not the case when $sep^{-/-}$ $securin^{-/-}$ oocytes were injected with mRNA to express both separase S1121A and securin (Fig. 3D,E), in accordance with inhibition of mutant separase by co-expressed securin. The presence of active separase allows us to conclude that

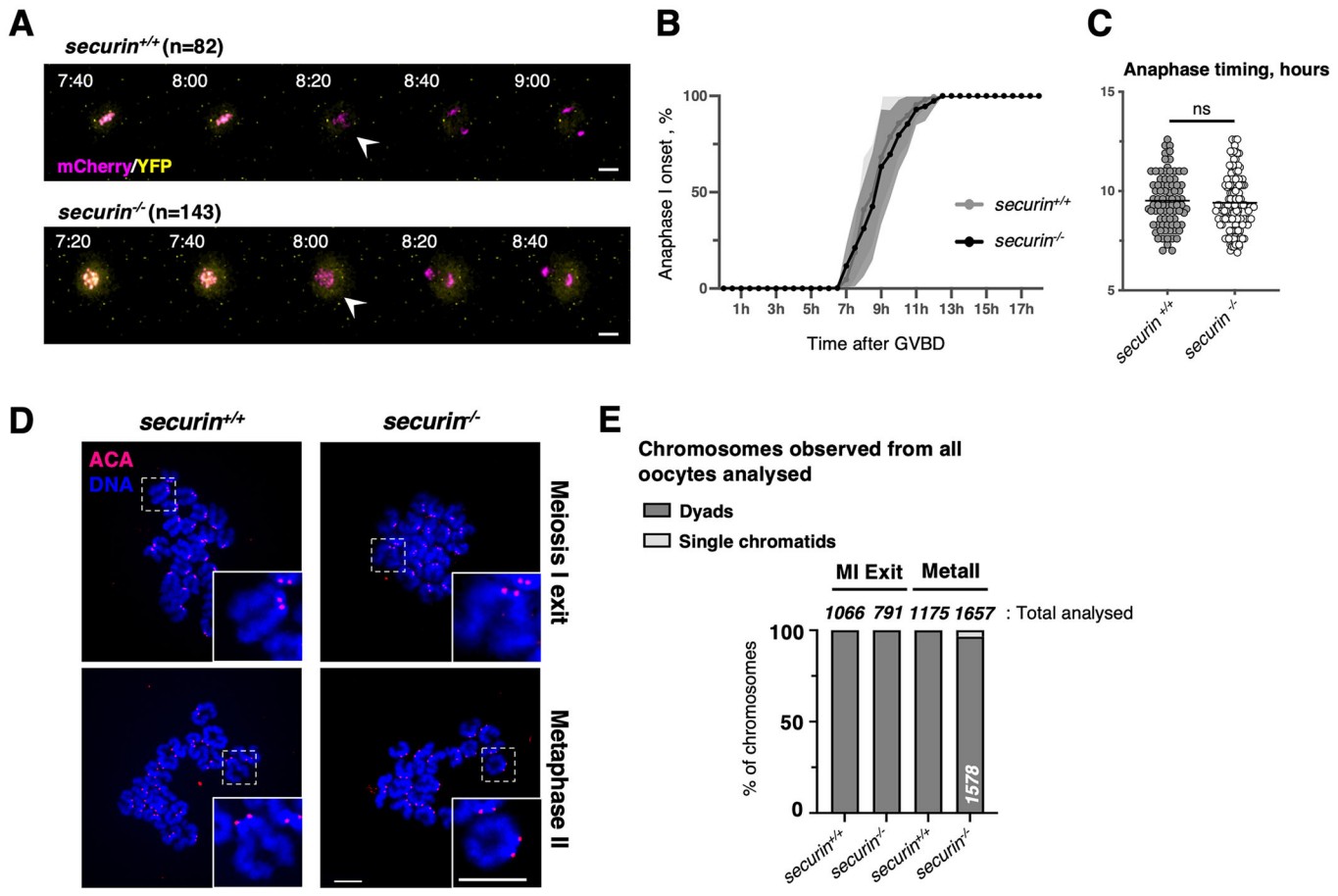

**Figure 2. Securin is not essential for separase inhibition in meiosis I.**

(A) Overlays of the YFP and mCherry channels of time-lapse microscopy acquisition, such as described in Fig. 1A, of wild-type (*securin*⁺/⁺) and securin knock-out (*securin*⁻/⁻) oocytes expressing the cleavage sensor. Time after GVBD is shown in hours:minutes (anaphase I onset is indicated with an arrowhead). n is the number of oocytes analyzed. Scale bar (white) represents 20 µm (related to Fig. EV2). (B) Monitoring anaphase I onset in the corresponding conditions of the selected time frames shown in (A). Hours after GVBD are indicated, and error bars are ±SD. (C) Statistical analysis of anaphase timing of *securin*⁺/⁺ and *securin*⁻/⁻ oocytes from (A). For each graph mean is shown, and ns indicates there is no significant difference according to Mann–Whitney *U* test. (D) *securin*⁺/⁺ and *securin*⁻/⁻ oocytes were fixed for chromosome spreads around 8 h after GVBD (Meiosis I Exit) and 20 h after GVBD (Metaphase II). Centromeres/kinetochores were stained with ACA (red) and chromosomes with DAPI (blue). A representative spread and magnification of one chromosome (white dashed line squares, insert at the bottom right corner) are shown for each condition. The total number of chromosomes analyzed is indicated in (E). Scale bars (white) represent 10 µm. (E) Frequency of chromosome categories observed at exit from meiosis I (8 h after GVBD) and at metaphase II arrest (20 h after GVBD), quantified from chromosome spreads in (D). The total number of chromosomes quantified for each condition, and of each category, is indicated. Data information: Results shown were obtained from at least three independent biological replicates. Source data are available online for this figure.

pericentromeric cohesin is not well protected at meiosis I resumption. Because cohesin gets cleaved so early, no real anaphase movement was observed (Fig. 3A; Movie EV2).

The absence of anaphase I movements in "separase-out-of-control" oocytes suggested that under these conditions, separase is activated and able to cleave all cohesin at a time when the APC/C is still inactive. To confirm that chromosomes and sister chromatids indeed separate in the absence of APC/C activation, we asked whether the SAC, which inhibits the APC/C, is active at the time of separase cleavage. In oocytes, it has been shown that the SAC is activated from GVBD until around 4 h after GVBD because of missing stable end-on attachments. The SAC gets progressively shut off as oocytes progress into metaphase I. This is revealed by staining for the SAC protein Mad2, which gets recruited to unattached kinetochores (Kitajima et al, 2011; Lane et al, 2012;

Wassmann et al, 2003). To establish whether separase cleaves pericentromeric and arm cohesin even though the APC/C is still inactive, we stained chromosome spreads for Mad2. Figure 3F,G shows that in both controls and "separase-out-of-control" oocytes, Mad2 is present at kinetochores in prometaphase I (3 h after GVBD). Strikingly, single sister chromatids in *sep*⁻/⁻ *securin*⁻/⁻ oocytes expressing separase S1121A showed a strong Mad2 signal at kinetochores. Mad2 staining was reduced in metaphase I, even in *sep*⁻/⁻ *securin*⁻/⁻ oocytes expressing separase S1121A, indicating that the SAC, which is only activated transiently in oocytes, had been turned off also in the presence of single sister chromatids. Treating "separase-out-of-control" oocytes with nocodazole at resumption of meiosis and at a concentration that prevents chromosome segregation in control oocytes, led to strong Mad2 recruitment. However, nocodazole treatment did not prevent

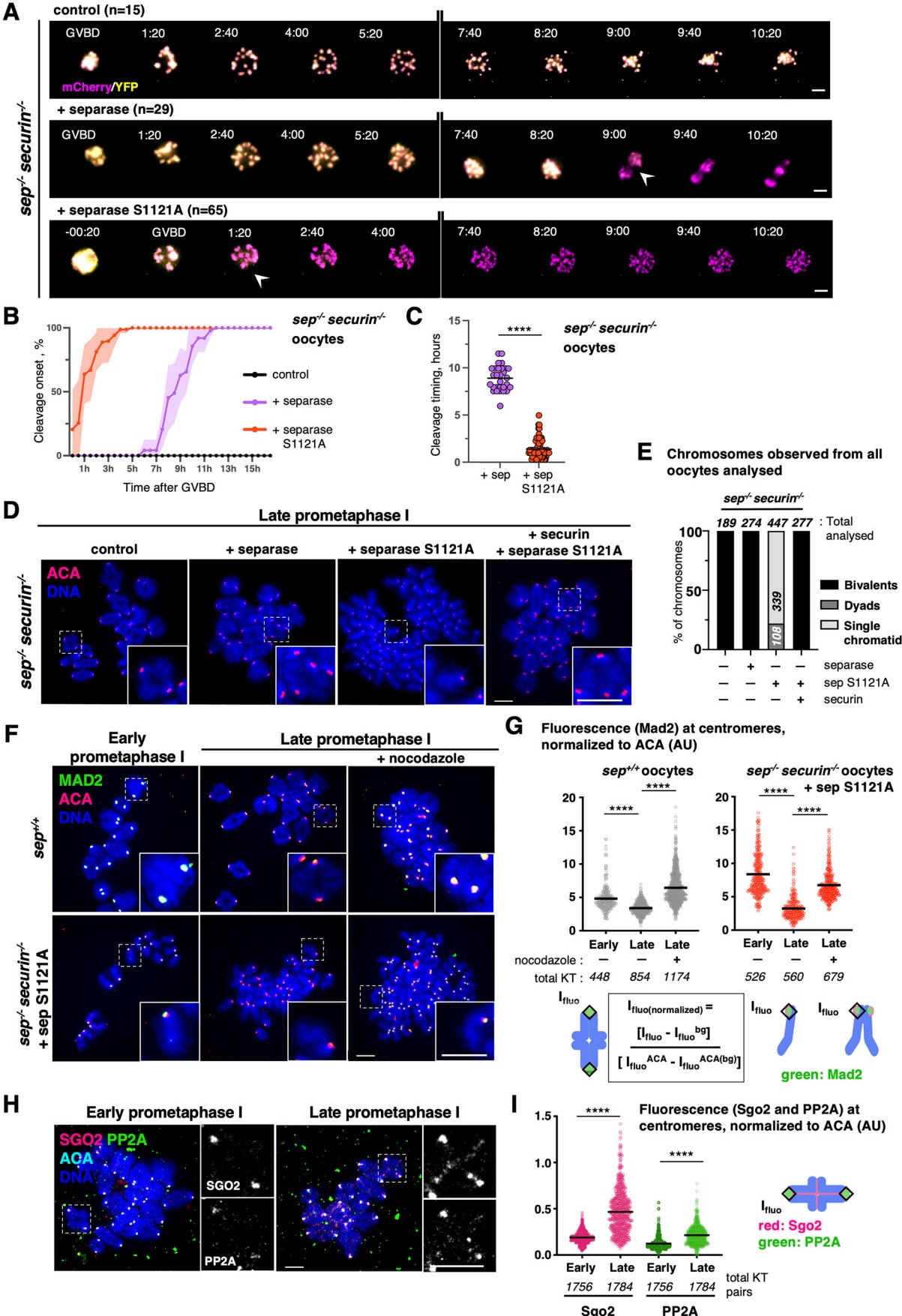

**Figure 3. Complete loss of separase control reveals absence of cohesin protection at meiosis resumption.**

(A) Overlays of the YFP and mCherry channels of time-lapse microscopy acquisition such as described for Fig. 1A, of $sep^{-/-}$ $securin^{-/-}$ oocytes expressing the cleavage sensor, from GVBD onwards. Time after GVBD is shown in hours:minutes (cleavage onset is indicated with an arrowhead). Oocytes have been co-injected with mRNAs encoding for separase or separase S1121A, where indicated. n is the number of oocytes analyzed. Scale bar (white) represents 20 µm (related to Fig. EV3; Movie EV2; Appendix Fig. S1). (B) Monitoring of the switch of color indicative of cleavage onset, in the corresponding conditions of the selected time frames shown in (A). Hours after GVBD are indicated, and error bars are ±SD. (C) Statistical analysis of cleavage timing in $sep^{-/-}$ $securin^{-/-}$ oocytes from (A). For each graph, mean is shown, and asterisks indicate significant difference (****$P = 9.6 \times 10^{-15}$) according to Mann–Whitney $U$ test. (D) $sep^{-/-}$ $securin^{-/-}$ oocytes expressing separase or separase S1121A, and additionally securin where indicated, were fixed for chromosome spreads around 5 h after GVBD (Late prometaphase I). Centromeres/kinetochores were stained with ACA (red) and chromosomes with DAPI (blue). A representative spread and magnification of one chromosome (white dashed line squares, insert at the bottom right corner) are shown for each condition. The total number of chromosomes analyzed is indicated in (E). Scale bars (white) represent 10 µm. (E) Frequency of chromosome categories observed around 5 h after GVBD (Late prometaphase I), quantified from chromosome spreads shown in (D). The total number of chromosomes quantified for each condition, and of each category, is indicated. (F) Not injected $sep^{+/+}$ (top) or $sep^{-/-}$ $securin^{-/-}$ (bottom) oocytes injected in GV with separase S1121A were fixed for chromosome spreads around 3 h after GVBD (Early prometaphase I) and around 6 h after GVBD (Late prometaphase I). Oocytes were treated with nocodazole at GVBD, where indicated. Chromosome spreads were stained for endogenous Mad2 (green), centromeres/kinetochores were stained with ACA (red) and chromosomes with DAPI (blue). A representative spread and magnification of one chromosome (white dashed line squares, insert at the bottom right corner) are shown for each condition. Scale bars (white) represent 10 µm. (G) Total Mad2 fluorescence intensities at centromeres/kinetochores were quantified from chromosome spreads in (F) in not injected $sep^{+/+}$ (right) and $sep^{-/-}$ $securin^{-/-}$ (left) oocytes expressing separase S1121A. Where indicated, oocytes were treated with nocodazole at GVBD. The signals measured (Ifluo) were corrected to background (bg) and normalized to ACA signals. On the graph, mean is shown, and asterisks indicate significant difference (on the left, early and late $sep^{+/+}$ oocytes: ****$P = 3.84 \times 10^{-9}$, late $sep^{+/+}$ oocytes +/− nocodazole: ****$P = 1.77 \times 10^{-121}$, on the right, early and late $sep^{-/-}$ $securin^{-/-}$ oocytes: ****$P = 8.16 \times 10^{-62}$, late $sep^{-/-}$ $securin^{-/-}$ oocytes +/− nocodazole: ****$P = 2.16 \times 10^{-103}$, according to Mann–Whitney $U$ test.) The number of kinetochore pairs (KT pairs) and kinetochore (KT) used for quantification is indicated. Below, scheme of Mad2 (green) signal measurement. AU, arbitrary units. (H) Wild-type oocytes were fixed for chromosome spreads at 1h 30 after GVBD (early prometaphase I) and 4 h after GVBD (late prometaphase I). Chromosome spreads were stained for endogenous Sgo2 (red), endogenous PP2A (green), centromeres/kinetochores were stained with ACA (cyan) and chromosomes with DAPI (blue). Representative spreads and magnifications of one chromosome (white dashed line squares) are shown for each condition, for Sgo2 (top) and PP2A (bottom) total staining. Scale bars (white) represent 10 µm. (I) Total Sgo2 (red) and total PP2A (green) fluorescence intensities at centromeres/kinetochores were quantified from chromosome spreads in (H). The signals measured (Ifluo) were corrected to background (bg) and normalized to ACA signals. On the graph, mean is shown, and asterisks indicate significant difference (****$P = 2.09 \times 10^{-176}$ for Sgo2, and ****$P = 4.77 \times 10^{-146}$ for PP2A, when comparing early and late prometaphase I) according to Mann–Whitney $U$ test. The number of kinetochore pairs (KT pairs) used for quantification is indicated. On the right, scheme of Sgo2 (red) and PP2A (green) signal measurements. AU arbitrary units. Data information: Results shown were obtained from at least three independent biological replicates. Source data are available online for this figure.

cohesin cleavage and sister chromatid separation, further indicating that it took place independently of APC/C activation (Figs. 3F,G and EV3B). We conclude that loss of securin and cyclin B1-mediated inhibition of separase leads to cleavage of all cohesin in an APC/C-independent manner upon meiosis I resumption.

Our results indicate that pericentromeric cohesin protection is absent when oocytes resume meiosis I. We hypothesized that separase activation happens so early in $sep^{-/-}$ $securin^{-/-}$ oocytes rescued with separase S1121A, that cohesin protection is not yet in place, leading to the observed phenotype. To further address this, we examined whether the cohesin protectors Sgo2 and PP2A are localized to the centromere region at all, early in meiosis I. Indeed, when checking for localization of Sgo2 and PP2A to the centromere region in early prometaphase I of wild-type oocytes, we found that their recruitment to centromeres increases with time and is significantly lower in early than in late prometaphase I (Fig. 3H,I). Pericentromeric cohesin protection is thus being gradually set up as oocytes progress through prometaphase of meiosis I. We conclude that the presence of active separase at meiosis resumption leads to the observed cleavage of all cohesin, on arms and at the centromere region, because pericentromeric cohesin is not yet fully protected at that time.

## Cyclin B1 is not the major inhibitor of separase in meiosis II

In a previous study, using injection experiments with blocking antibodies and transient knock-down approaches, securin and not cyclin B1-Cdk1 was proposed as the major separase inhibitor in oocyte meiosis II (Nabti et al, 2008). Using $sep^{-/-}$ oocytes, we set out to determine whether cyclin B1 on its own is indeed not required for

separase inhibition in meiosis II. As we have shown previously, $sep^{-/-}$ oocytes cannot separate chromosomes in meiosis I, however, as chromosomes are correctly attached, the SAC is satisfied and the APC/C is activated (Kudo et al, 2006). These oocytes progress through meiosis I into meiosis II, with the degradation and reaccumulation of APC/C substrates. Hence, $sep^{-/-}$ oocytes that have reached meiosis II can be injected with mRNA coding for wild-type separase or S1121A separase to ask whether chromosome separation is observed, indicating separase activation (Gryaznova et al, 2021). Thus, this experimental tool allows us to address whether separase inhibition by cyclin B1 becomes important during the prolonged metaphase II arrest that oocytes are subjected to when they are awaiting fertilization (Fig. 4A).

As expected, injection of wild-type separase into control or $sep^{-/-}$ oocytes did not lead to chromosome or sister chromatid separation during metaphase II arrest, because separase remained correctly inhibited (Fig. 4B–E). Importantly, the same was true when separase S1121A was expressed, showing that phosphorylation of S1121 by cyclin B1-Cdk1 is not essential, or not the sole mechanism, required to keep separase under control during the metaphase II arrest (Fig. 4B–E). Thus, cyclin B1 is the sole mechanism inhibiting separase at the transition from meiosis I into meiosis II (Fig. 1G,H), but once oocytes have reached metaphase II, loss of cyclin B1 inhibition alone was not enough to activate separase. Mimicking fertilization through strontium treatment induced anaphase II onset in control and $sep^{-/-}$ oocytes (Figs. 4C and EV4A). As we have shown previously, $sep^{-/-}$ oocytes segregate chromosomes into dyads and not sister chromatids (Fig. EV4B), because sister kinetochore individualization -which depends on separase activity and takes place in anaphase I- has not occurred (Gryaznova et al, 2021). Crucially, both wild-type and S1121A separase became activated only after mimicking fertilization

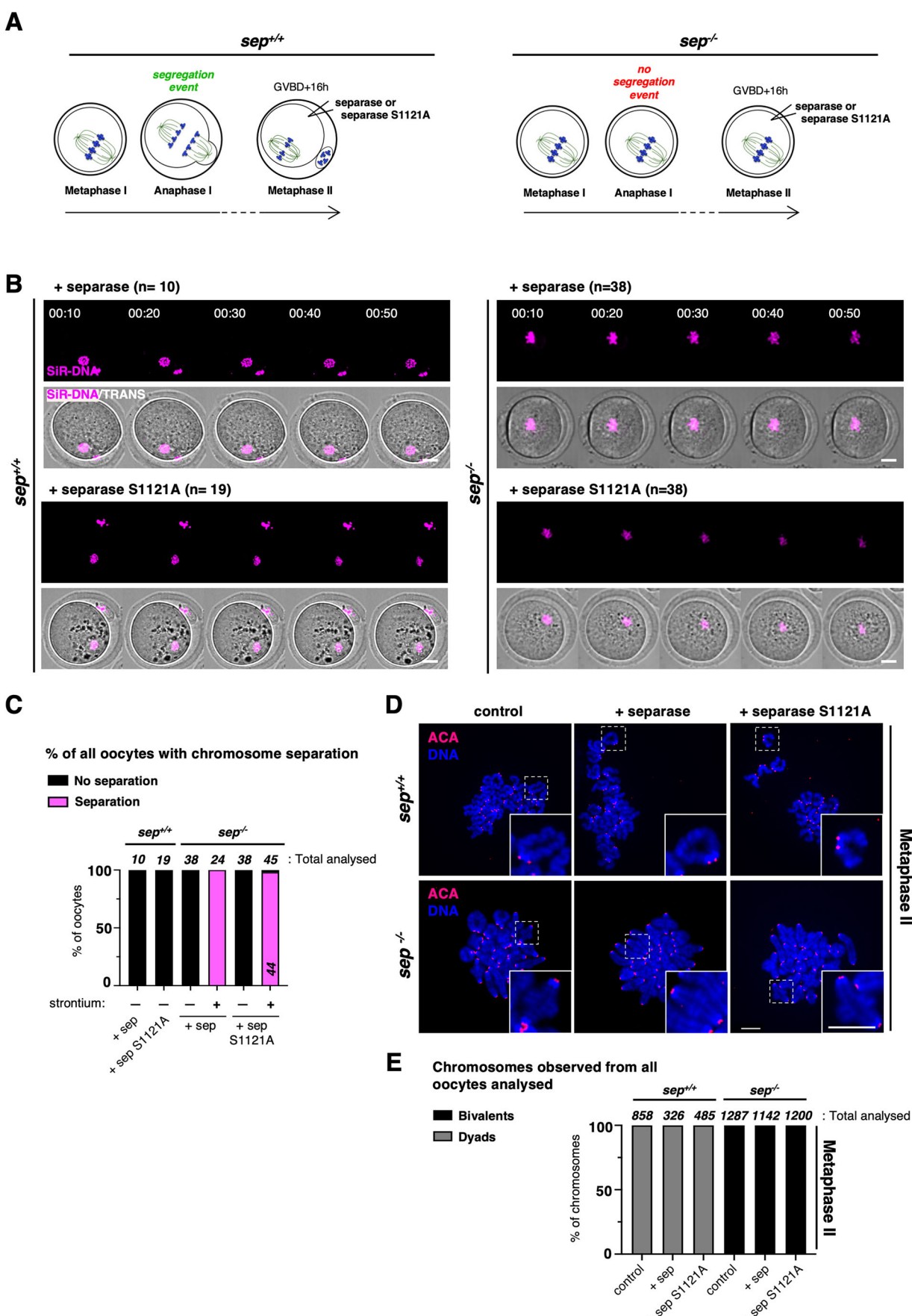

Figure 4.   Inhibition of separase by cyclin B1 or securin alone is not essential in meiosis II.

(A) Illustrative scheme of the experimental setup used. Oocytes of the indicated genotype were matured in vitro until metaphase II. $sep^{+/+}$ correspond to $separase^{LoxP/LoxP}$. Of note, $sep^{-/-}$ oocytes progress into metaphase II, even though no chromosome segregation occurs in meiosis I (Kudo et al, 2006). Once arrested at metaphase II, oocytes were injected with mRNAs encoding for separase or separase S1121A. Post injections, oocytes were either subjected to live imaging to track chromosome segregation (B), or chromosome spreads were performed to verify cohesin cleavage (D). (B) Live imaging of $sep^{+/+}$ (left) or $sep^{-/-}$ (right) oocytes undergoing meiosis II (from metaphase II onwards) and expressing separase or separase S1121A. For all conditions, oocytes were injected around 16 h after GVBD (arrested in metaphase II) and were subjected to live imaging around 18 h after GVBD. On top, the magenta channel to visualize chromosomes, and below overlay with TRANS channel image. Time after start of the movie is shown in hours:minutes. Prior to acquisition, oocytes were preincubated in culture media containing SiR-DNA to visualize chromosomes. Efficient expression of the separase constructs was verified by inducing anaphase II chemically with strontium (related to Fig. EV4A). n is the number of oocytes analyzed. Scale bar (white) represents 20 μm. (C) Frequency of DNA separation observed in metaphase II-arrested oocytes, indicating separase activation, in (B). The total number of oocytes analyzed for each condition, and of each category, is indicated. sep: separase (related to Fig. EV4A). (D) $sep^{+/+}$ (top) or $sep^{-/-}$ (bottom) oocytes injected around 16 h after GVBD (arrested in metaphase II) with the indicated constructs, were fixed for chromosome spreads around 22 h after GVBD (Metaphase II). Centromeres/kinetochores were stained with ACA (red) and chromosomes with DAPI (blue). A representative spread and magnification of one chromosome (white dashed line squares, insert at the bottom right corner) are shown for each condition. The total number of chromosomes analyzed is indicated in (E). Scale bars (white) represent 10 μm (related to Fig. EV4B). (E) Frequency of chromosome categories observed at metaphase II arrest (22 h after GVBD), quantified from chromosome spreads shown in (D). The total number of chromosomes quantified for each condition, is indicated. sep separase. Data information: Results shown were obtained from at least three independent biological replicates. Source data are available online for this figure.

and not before, indicating that the exogenously expressed separase was functional and activity was correctly controlled.

## Securin is not an essential inhibitor of separase in meiosis II

As shown before, securin is not required during the transition from meiosis I into meiosis II to keep separase under control, and $securin^{-/-}$ oocytes progress into metaphase II where they remain arrested to await fertilization, without showing significant precocious sister chromatid segregation during the arrest (Fig. 2D,E). To create conditions where separase is only inhibited by cyclin B1 and not securin, we used double knock-out $sep^{-/-}$ $securin^{-/-}$ oocytes that were allowed to progress into metaphase II. In metaphase II, we expressed wild-type separase, which can still be inhibited by cyclin B1 (Fig. 5A). When wild-type separase was expressed in $sep^{-/-}$ $securin^{-/-}$ metaphase II oocytes, the absence of securin in metaphase II did not cause chromosome or sister chromatid separation (Fig. 5B, top right, C,D, bottom, E). This demonstrates that on the contrary to (Nabti et al, 2008), cyclin B1 on its own is sufficient to inhibit separase, and securin is not the major separase inhibitor in metaphase II. We asked whether the amount of exogenous separase was sufficient for cohesin cleavage in the metaphase II rescue experiment. Inducing anaphase II onset by strontium treatment resulted in the timely segregation of chromosomes, indicating that enough active separase was expressed (Figs. 5C and EV4A).

## Meiosis II without separase inhibition

As before in meiosis I, we asked whether loss of both cyclin B1 and securin inhibition of separase leads to constitutively active separase, independently of cell cycle progression into anaphase II. If not, a third inhibitor such as Sgo2-Mad2 may contribute to separase inhibition in meiosis II and keep separase inactive (Hellmuth et al, 2020). $sep^{-/-}$ $securin^{-/-}$ oocytes in metaphase II were rescued with separase S1121A to obtain active separase, decoupled from cell cycle progression, in meiosis II. Indeed, such as in meiosis I, separase S1121A is not inhibited and cleaves cohesin even though oocytes have not been fertilized or activated with strontium (Figs. 5A,B, bottom right, C,D, bottom right, E and EV4A).

As we have shown previously, pericentromeric cohesin is still protected in metaphase II $sep^{-/-}$ oocytes because they did not undergo sister kinetochore individualization (Gryaznova et al, 2021). Accordingly, $sep^{-/-}$ $securin^{-/-}$ oocytes expressing S1121A were only separating chromosomes into dyads, and hardly any sister chromatids were observed. The resulting dyads had individualized kinetochores, such as previously observed when wild-type separase was expressed from metaphase II onwards in $sep^{-/-}$ oocytes treated with strontium to undergo anaphase II (Fig. 5D,E). This data is in agreement with our earlier findings that sister kinetochore individualization mediating deprotection has to take place in the previous division for centromeric cohesin cleavage, indicating that individualization and centromeric cohesin cleavage must be temporally separated (Gryaznova et al, 2021). Importantly, prolonged incubation of $sep^{-/-}$ $securin^{-/-}$ oocytes expressing S1121A in metaphase II did not lead to a significant increase in the presence of single sister chromatids (Fig. EV4C). This shows that missing inhibition of separase (or extended presence of active separase) did not lead to the cleavage of pericentromeric cohesin, even after an extended period of time. Thus, our experimental setting allowed us to obtain constitutively active separase in metaphase II in the presence of pericentromeric cohesin protection, and showed that cohesin protection is not overridden with time.

## Metaphase II without separase inhibition reveals the absence of efficient cohesin protection

Having determined how separase inhibition can be abolished in meiosis II allowed us to attempt answering the long-standing question of whether pericentromeric cohesin protection is still present throughout the extended metaphase II arrest in oocytes that harbor the usual dyads. We used $securin^{-/-}$ oocytes, which progress normally through meiosis I with the segregation of chromosomes into dyads and kinetochore individualization due to the activity of endogenous separase (Fig. 2). S1121A separase was expressed in addition to endogenous separase in metaphase II $securin^{-/-}$ oocytes (Fig. 6A). Importantly, we found that expression of S1121A separase (which is inhibited neither by cyclin B1 nor securin in this context) led to immediate sister chromatid separation of nearly all dyads (Fig. 6B,C). This result indicates that pericentromeric

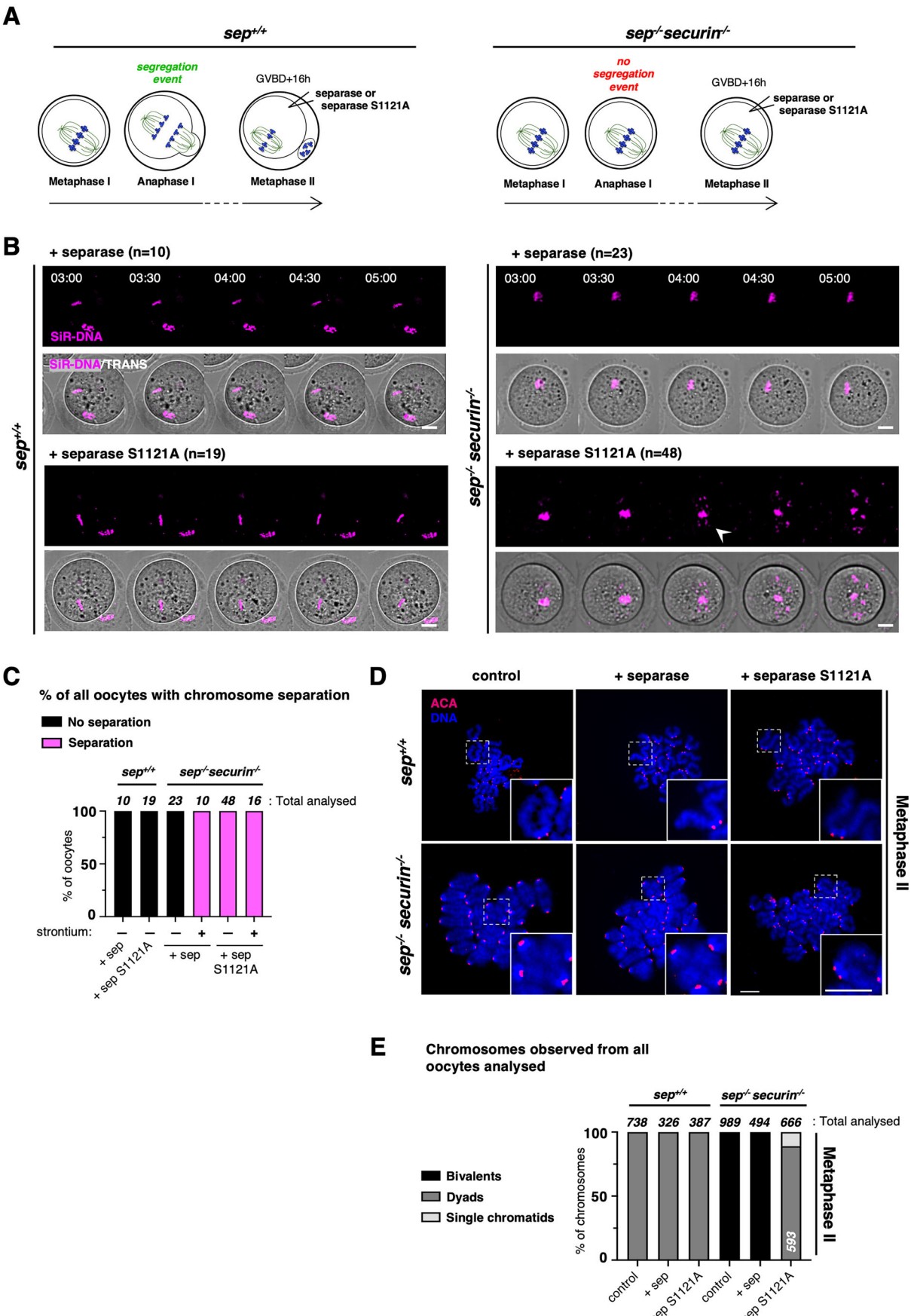

**Figure 5.   Loss of separase control by interfering with both cyclin B1 and securin in meiosis II.**

(A) Illustrative scheme of the experimental setup: $sep^{+/+}$ ($separase^{LoxP/LoxP}$) and $sep^{-/-}securin^{-/-}$ oocytes were matured in vitro until metaphase II and injected with mRNAs encoding for separase or separase S1121A. Post injections, oocytes were subjected to live imaging to track either chromosome segregation (B), or chromosome spreads were performed to verify cohesin cleavage (D). (B) Live imaging of $sep^{+/+}$ (left) and $sep^{-/-}securin^{-/-}$ (right) oocytes expressing separase or separase S1121A, where indicated, such as described in (A). For all conditions, oocytes were injected around 16 h after GVBD (arrested in metaphase II) and were subjected to live imaging around 18 h after GVBD. Time after start of the movie is shown in hours:minutes (separation of chromosomes is indicated with an arrowhead). Prior to acquisition, oocytes were preincubated in culture media containing SiR-DNA to visualize chromosomes. Efficient expression of the separase constructs was verified by inducing anaphase II chemically with strontium. n is the number of oocytes analyzed. Scale bar (white) represents 20 μm (related to Fig. EV4A and Movie EV3). (C) Frequency of DNA separation observed in metaphase II-arrested oocytes, indicating separase activation, in the corresponding conditions of the selected time frames shown in (B). The total number of oocytes analyzed for each condition, and of each category, is indicated. sep: separase (related to Fig. EV4A). (D) $sep^{+/+}$ (top) and $sep^{-/-}securin^{-/-}$ (bottom) oocytes were injected around 16 h after GVBD (arrested in metaphase II) with the indicated constructs, and were fixed for chromosome spreads around 22 h after GVBD (Metaphase II). Centromeres/kinetochores were stained with ACA (red) and chromosomes with DAPI (blue). A representative spread and magnification of one chromosome (white dashed line squares, insert at the bottom right corner) are shown for each condition. The total number of chromosomes analyzed is indicated in (E). Scale bars (white) represent 10 μm (related to Fig. EV4C). (E) Frequency of chromosome categories observed at metaphase II arrest (22 h after GVBD), quantified from chromosome spreads shown in (D). The total number of chromosomes quantified for each condition, and of each category, is indicated. sep separase. Data information: Results shown were obtained from at least three independent biological replicates. Source data are available online for this figure.

cohesin protection is largely absent in metaphase II when kinetochore individualization has previously taken place. Even though Sgo2 and PP2A are present at metaphase II kinetochores and centromeres (Chambon et al, 2013; Gryaznova et al, 2021; Mengoli et al, 2021), they do not confer efficient pericentromeric cohesin protection.

## APC/C activation is not required for sister chromatid separation in meiosis II

In budding yeast metaphase II, deprotection of pericentromeric cohesin was shown to depend on APC/C activation, interfering with Sgo1-PP2A protective function (Jonak et al, 2017; Mengoli et al, 2021). We were wondering whether APC/C activation was equally required for sister chromatid segregation in metaphase II oocytes. In other words, was the cleavage of centromeric cohesin upon S1121A separase expression in $securin^{-/-}$ oocytes due to spontaneous anaphase II onset, or did it take place independently of cell cycle progression? In metaphase II-arrested oocytes, all kinetochores are attached, thus we could not use SAC satisfaction as a readout for APC/C activation, such as in meiosis I. Hence, to address this question, and without interfering with separase activity, we followed degradation of a cyclin B1 mutant, which is devoid of kinase activity due to mutations in the MRAIL motif (required for correct binding to Cdk1, MRL mutant (Jeffrey et al, 1995; Schulman et al, 1998)) and which carries a fluorescent tag. As expected, $securin^{-/-}$ and $sep^{-/-}$ $securin^{-/-}$ oocytes that were activated with strontium showed degradation of cyclin B1 MRL-GFP by live imaging. However, $securin^{-/-}$ and $sep^{-/-}$ $securin^{-/-}$ oocytes injected with S1121A separase and not subjected to activation, separated sister chromatids or chromosomes, respectively, but did not degrade cyclin B1 MRL-GFP. (Fig. 6D–G). No proper anaphase movements, nor any attempts of second PB extrusion were observed, further showing that oocytes are still cell cycle arrested (Fig. EV5; Movie EV3). Thus, we conclude that cleavage of pericentromeric cohesin and sister chromatid segregation in oocyte meiosis II only depends on prior kinetochore individualization, probably due to centromeric cohesin cleavage, and separase activation. Constitutively active separase can cleave pericentromeric cohesin without APC/C activation.

## Discussion

Separase cleaves the kleisin subunit of the cohesin complex, and its inappropriate activation leads to cohesin loss with weakened physical connections holding sister chromatids together. In the worst case, sister chromatids will separate precociously, leading to both mitotic and meiotic aneuploidies. Separase could thus be categorized as a "dangerous" enzyme requiring perfect control. This is reflected, for example, by the fact that intracellular securin homeostasis is important for maintaining integrity of pituitary cell adenoma formation (Chesnokova et al, 2008). In mitosis, the time when separase could inappropriately cleave mitotic Scc1 and thus, weaken cohesion, may be rather limited, as before entry into mitosis, separase is localized to the cytoplasm and physically separated from cohesin. Then, until metaphase-to-anaphase transition, separase is kept under control by three independent mechanisms: (1) securin, which functions as a pseudo-substrate inhibitor and at the same time, is required for full separase activity through its chaperone activity; (2) cyclin B1-Cdk1, which phosphorylates separase and creates a pocket binding site to bind separase, resulting in inhibition of separase by cyclin B1, but equally, of cyclin B1-Cdk1's kinase activity by separase; and (3) Sgo2-Mad2, proposed to bind separase and inhibit its cleavage activity in a pseudo-substrate dependent manner. Exit from the mitotic cell cycle with the reformation of a nuclear envelope and nuclear exclusion of separase protects cohesin from being cleaved until the subsequent mitosis (Wassmann, 2022; Yu et al, 2023).

### Separase inhibition until metaphase I

Our study was motivated by the quest to determine when, during oocyte meiosis I and II, pericentromeric cohesin is protected from cleavage by separase. For this, it was necessary to generate oocytes containing separase that is constitutively active by removing its inhibitors. We found that in oocytes, either securin or cyclin B1 is sufficient to maintain separase under control until metaphase I, confirming previous studies (Chiang et al, 2011; Wetherall et al, 2025). At the transition from meiosis I into meiosis II, cyclin B1 plays a major role in separase inhibition, suggesting that securin-dependent inhibition is not important at this cell cycle stage specific to meiosis. This premise fits well with the fact that cyclin B1

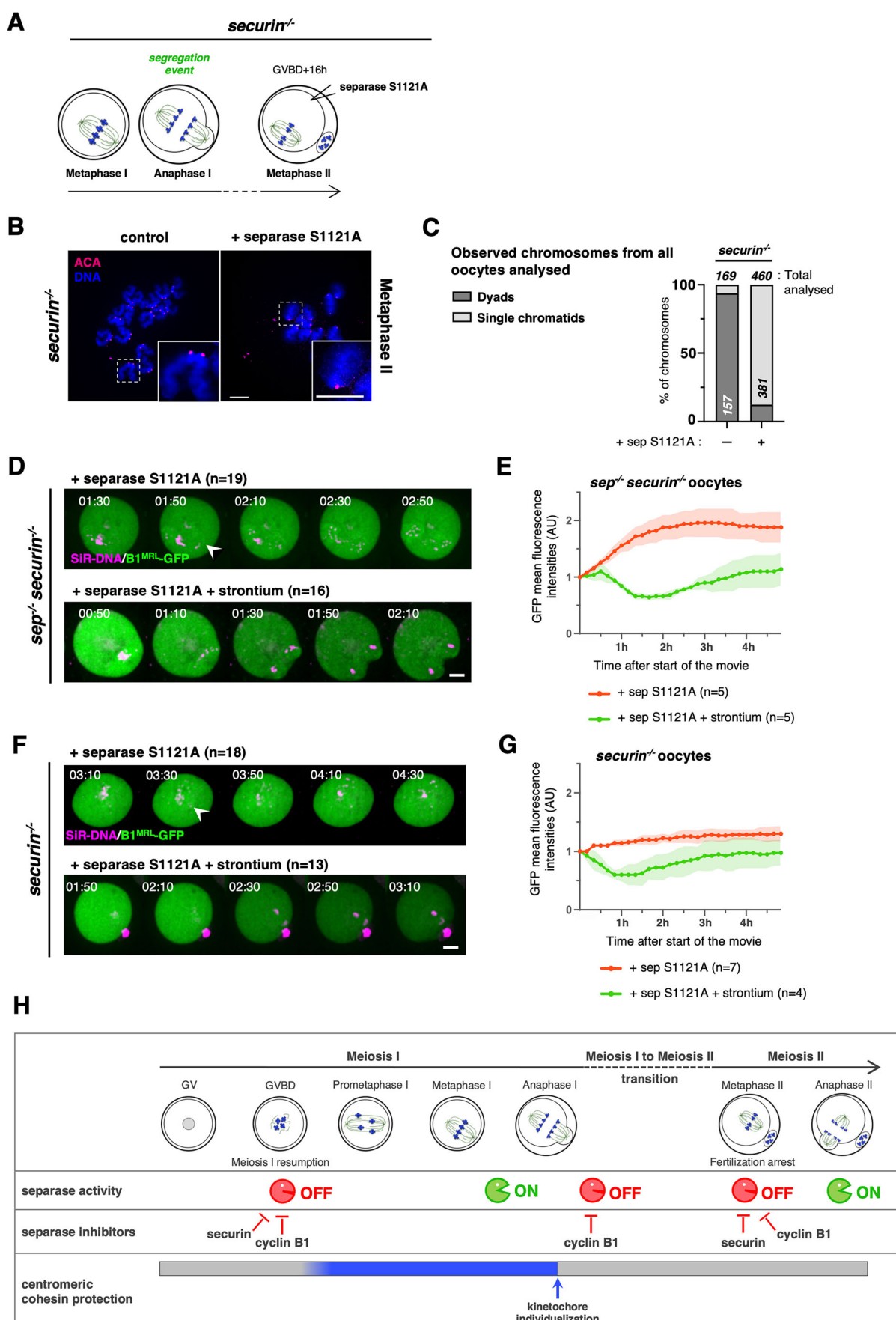

© The Author(s)

**Figure 6. Loss of separase inhibition by cyclin B1 and securin in meiosis II reveals absence of efficient cohesin protection.**

(A) Illustrative scheme of the experimental setup: securin$^{-/-}$ oocytes were matured in vitro until metaphase II and injected with mRNAs encoding for separase S1121A around 16 h after GVBD, such as described in Fig. 4A. (B) securin$^{-/-}$ oocytes expressing S1121A separase, where indicated, were fixed for chromosome spreads around 22 h after GVBD (Metaphase II). Chromosome spreads were stained for centromeres/kinetochores were stained with ACA (red) and chromosomes with DAPI (blue). A representative spread and magnification of one chromosome (white dashed line squares, insert at the bottom right corner) are shown for each condition. The total number of chromosomes analyzed is indicated in (C). Scale bars (white) represent 10 µm. (C) Frequency of chromosome categories observed at metaphase II arrest (22 h after GVBD), quantified from chromosome spreads shown in (B). The total number of chromosomes quantified for each condition, and of each category, is indicated. (D) Overlays of live imaging of sep$^{-/-}$securin$^{-/-}$ double knock-out oocytes expressing separase S1121A and the GFP-cyclin B1 MRL mutant. Oocytes were injected with the constructs around 16 h after GVBD, and activated with strontium, where indicated. Oocytes were then subjected to live imaging around 18 h after GVBD, corresponding to 5 min after activation with strontium. Time after start of the movie is shown in hours:minutes (separation of chromosomes is indicated with an arrowhead) (related to Fig. EV5). (E) Quantification of GFP-cyclin B1 MRL in the corresponding conditions of the selected time frames shown in (D). Hours after the start of the movie are indicated, error bars are ±SD, and n is number oocytes used for quantification. Shown, quantification of one representative replicate. (F) Overlays of live imaging of securin$^{-/-}$ knock-out oocytes activated with strontium, where indicated, expressing separase S1121A and the GFP-cyclin B1 MRL mutant such as described in (D). Time after start of the movie is shown in hours:minutes (separation of chromosomes is indicated with an arrowhead) (related to Fig. EV5). (G) Quantification of GFP-cyclin B1 MRL in the corresponding conditions of the selected time frames shown in (F). Hours after the start of the movie are indicated, error bars are ±SD, and n is number oocytes used for quantification. Shown, quantification of one representative replicate. (H) Model of how separase activity and centromeric cohesin protection are coordinated during meiotic cell cycle progression in oocytes. See text for details. Data information: Results shown were obtained from at least three independent biological replicates. Source data are available online for this figure.

degradation occurs in a delayed manner compared to securin, and that some cyclin B1 protein can still be detected at exit from meiosis I, even though Cdk1 kinase activity is undetectable (Karasu et al, 2019; Levasseur et al, 2019; Niault et al, 2007; Thomas et al, 2021). The complete loss of separase control shows that Sgo2-Mad2 is not essential in oocyte meiosis for separase control, and this has been recently confirmed by an independent study (Wetherall et al, 2025). Sgo2-Mad2 cannot substitute for loss of securin and cyclin B1 in this respect, and if Sgo2-Mad2 has a role, it is likely redundant with the two other inhibitors. However, we cannot exclude that Sgo2-Mad2 may become important once protection has been set up. In early meiosis I, Sgo2 may not yet have been recruited to sufficiently high levels yet- not only for protection, but also to inhibit separase. But to address the role of Sgo2-Mad2 convincingly, it will be necessary to invalidate Sgo2 or Mad2 specifically in metaphase I and not before, and find ways to separate a potential role in separase inhibition from other essential functions both proteins play in oocyte meiosis, such as cohesin protection, chromosome alignment and SAC control.

## Protection of pericentromeric cohesin is not yet properly set up when oocytes resume meiosis I

All cohesin is cleaved at meiosis I resumption, when both securin and cyclin B1 as inhibitors are absent, implying that proper pericentromeric cohesin protection is set up rather late during meiosis I (Fig. 6H). This correlates with our results showing that Sgo2 and PP2A are reduced at the centromere region in early prometaphase I. In this respect, it is interesting to note that oocytes devoid of spindle assembly checkpoint components and thus undergoing a strongly accelerated meiosis I with anaphase I around 4–5 h after GVBD, missegregate chromosomes, but not sister chromatids. (One exception is the checkpoint kinase Mps1, which also contributes to Sgo2 localization for protection.) (El Yakoubi et al, 2017; McGuinness et al, 2009; Touati et al, 2015). Hence, protection of cohesin is functional at least 4–5 h after meiosis resumption, and this fits well with our findings that Sgo2 and PP2A levels around centromeres are much higher at that time compared to early prometaphase I. Thus, we identified a critical moment at the start of oocyte maturation that may contribute to missegregations and thus the generation of aneuploidies, when tight separase control is perturbed.

## Transition from meiosis I into meiosis II

Another critical moment in oocyte maturation is the transition from meiosis I into meiosis II, because no nuclear envelope -to separate separase from Rec8- is reformed during this cell cycle transition. Additionally, separase inhibition depends only on cyclin B1 and not securin. The fact that Sgo2-PP2A become strongly reduced at the centromeric region as oocytes progress from meiosis I into meiosis II prompted us in the past to ask whether an additional mechanism prevents separase from accessing pericentromeric cohesin. Indeed, we found that sister kinetochore individualization is required for deprotection and cleavage of pericentromeric cohesin (Gryaznova et al, 2021). However, this individualization takes place in anaphase I, and thus well before reaccumulation of securin and cyclin B1. It remains enigmatic how pericentromeric cohesin cleavage is avoided once sister kinetochore individualization has taken place, and when cyclin B1 and securin are still absent to re-inhibit separase. As an additional control to avoid that separase cleaves pericentromeric cohesin once cyclin B1 is degraded, the kinase phosphorylating Rec8 for cleavage in meiosis II may be localized away from Rec8. Indeed, Aurora B/C, the kinase(s) promoting Rec8 phosphorylation in meiosis I, is/are part of the chromosomal passenger complex and thus localized to the spindle midzone (Nguyen and Schindler, 2017; Nikalayevich et al, 2022). Also, it has been shown that the SUMO pathway and the E3 ligase PIAS are required for pericentromeric cohesin maintenance after anaphase I, however, the molecular targets are currently unknown (Ding et al, 2018).

It has been proposed that cyclin B1-Cdk1 activity only partially recedes between meiosis I and meiosis II, allowing certain substrates to remain phosphorylated during the transition and thus, preventing exit into interphase. It remains to be seen whether residual cyclin B1-Cdk1 activity must be retained to maintain separase inactive. We have previously found that some cyclin B1 protein is still present in oocytes exiting meiosis I, even though kinase activity was undetectable (Karasu et al, 2019). Cyclin B1-Cdk1 may thus inhibit separase during the short time window when the APC/C is active and cyclin B1-Cdk1-associated activity is undetectable, before reaccumulation of cyclin B1 in prometaphase of meiosis II. This hypothesis would be consistent with our result

that precocious sister chromatid segregation is observed in oocytes that progress into meiosis II without cyclin B1-dependent inhibition of separase (Fig. 1G,H). However, more experiments are needed to confirm or invalidate this hypothesis.

## Cohesin is already deprotected during the oocyte metaphase II arrest

In budding yeast, artificial activation of separase in metaphase II-arrested cells (due to conditional inhibition of the APC/C activator Cdc20) does not lead to pericentromeric cohesin removal. Additional removal of Sgo1 was necessary to deprotect Rec8 and induce sister chromatid segregation (Mengoli et al, 2021). Thus, active separase is not sufficient in budding yeast metaphase II to cleave pericentromeric Rec8, because protection mediated by Sgo1 is still present and also needs to be removed. This is in contrast with our findings here in mouse oocytes, where elimination of both cyclin B1- and securin-dependent inhibition of separase led to immediate separation of sister chromatids in metaphase II. We expected that APC/C activation would still be required to remove cohesin protection at this time, such as in budding yeast. However, we found here that cleavage of pericentromeric cohesin took place without APC/C activation, or visible anaphase movements. Hence, Sgo2 and PP2A localized to the centromere region were apparently unable to protect pericentromeric cohesin efficiently in the presence of active separase at this stage of the cell cycle. Crucially, Sgo2 and PP2A are also involved in other functions at the kinetochore/centromere in oocytes (Rattani et al, 2013). Therefore, we conclude that their sole presence does not indicate that protection is indeed functional, at least in meiosis II. The only condition for cleavage of pericentromeric cohesin in meiosis II in oocytes was previous bivalent segregation. This finding is in accordance with our model that sister kinetochore individualization of dyads occurring at meiosis I exit is essential for separase to access pericentromeric cohesin, but only in the following division, meiosis II (Gryaznova et al, 2021). We speculate that cohesin protection is still present in budding yeast metaphase II, as in this model system, only Pds1 (securin) functions as a separase inhibitor (Kamenz and Hauf, 2017; Mengoli et al, 2021). Yeast meiosis may thus additionally rely on protection of pericentromeric cohesin to avoid precocious sister chromatid segregation. In mouse oocytes, inhibition of separase by securin and cyclin B1-Cdk1, and distinct localization of the kinase phosphorylating Rec8 likely substitute for protection in meiosis II.

In conclusion, our results show that either of the main known inhibitors of vertebrate separase—namely securin or cyclin B1-Cdk1—is sufficient to maintain separase under control, except at the transition from meiosis I into meiosis II, which depends mainly on cyclin B1-Cdk1. Furthermore, we found that tight pericentromeric cohesin protection is in place only during a relatively short time window during meiosis I (Fig. 6H). Our study thus elucidates critical moments during oocyte maturation when aneuploidies may arise in vertebrate meiosis. Overexpression of separase or prolonged presence of active separase has been shown to lead to weakening of cohesion, even when protected, and the generation of aneuploidies, in both mitosis and meiosis (Chiang et al, 2011; Mukherjee et al, 2014; Zhang et al, 2008). This indicates that with time, also protected cohesin will be cleaved in the presence of active separase. It remains intriguing that oocyte meiosis does not depend on additional mechanisms to avoid untimely cohesin removal throughout meiotic maturation and during the extended metaphase II arrest.

## Methods

### Reagents and tools table

| Reagent/resource | Reference or source | Identifier or catalog number |
|---|---|---|
| **Experimental models** | | |
| C57Bl6/J (M. musculus) | Janvier Labs | C57BL/6-Jrj |
| Swiss (M. musculus) | Janvier Labs | RjOrl:SWISS (CD-1) |
| **Recombinant DNA** | | |
| pRN3-H2B-mCherry-hScc1(107-268)-YFP | This study | Nikalayevich et al (2022) |
| pRN3-mseparase | Kudo et al (2006) | |
| pRN3-mseparase S1121A | Touati et al (2012) | |
| pRN3-msecurin | Touati et al (2012) | |
| pRN3-human Cyclin B1(MRL)-mGFP | This study | |
| **Antibodies** | | |
| Human anti-centromere protein (ACA) | Cliniciences | 15-234 |
| Human CREST serum | Immunovision | HCT-100 |
| Rabbit anti-Rec8 | Gryaznova et al (2021) | |
| Rabbit anti-Sgo2 | This study | |
| Mouse anti-PP2AC-1D6-Alexa Fluor 488 | Sigma-Aldrich | 05-421-AF488 |
| Rabbit anti-Mad2 | Wassmann and Benezra, (1998) | |
| Donkey anti-human CY3 | Jackson Immuno Research | 709-166-149 |
| Donkey anti-human Alexa Fluor 647 | Jackson Immuno Research | 709-606-149 |
| Donkey anti-rabbit CY3 | Jackson Immuno Research | 711-166-152 |
| Donkey anti-rabbit Alexa Fluor 488 | Jackson Immuno Research | 711-546-152 |
| Donkey anti-mouse Alexa Fluor 488 | Jackson Immuno Research | 715-546-150 |
| **Chemicals, enzymes, and other reagents** | | |
| dibutyryl cyclic AMP | Sigma-Aldrich | D0260 |
| Embryo culture oil | FUJIFILM Irvine Scientific | 9305 |

| Reagent/resource | Reference or source | Identifier or catalog number |
|---|---|---|
| Strontium chloride hexahydrate | Sigma-Aldrich | 204463 |
| Paraformaldehyde | Sigma-Aldrich | 441244 |
| Triton X-100 | Sigma-Aldrich | T8787 |
| Dithiothreitol (DTT) | Sigma-Aldrich | D9779 |
| Bovine Serum Albumin (BSA) | Sigma-Aldrich | A4378 |
| Vectashield DAPI | EUROBIO Scientific | H-1200 |
| SiR-DNA | Spirochrome | SC007 |
| Nocodazole | Sigma-Aldrich | M1404 |
| **Software** | | |
| Metamorph 7.10.3 | Metamoprh | https://support.moleculardevices.com/s/article/MetaMorph-Software-installation-files |
| Fiji 5.1.0 | Fiji | https://imagej.net/software/fiji/ |
| PRISM 10 | GraphPad | https://www.graphpad.com/scientific-software/prism/ |
| **Other** | | |
| mMessage mMachine T3 kit | Invitrogen | AM1348 |
| RNeasy Mini Kit 5 | Qiagen | 74104 |

## Animals

Mice were kept in an enriched environment with ad libitum food and water access in the conventional animal facility of UMR7622 (authorization C75-05-13) and UMR7592 (authorization C75-13-17), according to current French guidelines. Housing was done under a 12-h light/12-h dark cycle according to the Federation of European Laboratory Science Associations (FELASA). All experiments were subject to ethical review and approved by the French Ministry of Higher Education and Research (authorization n° B-75-0513), in accordance with national guidelines and application of the "3 Rs" rule (Licence 5330). The number of mice used was kept as low as possible but high enough to obtain statistically significant results, in agreement with the standards used in the field. Females for experiments were either house-bred in the animal facility of UMR7592 (genetic knock-out and littermate control mice), or purchased at 7 weeks of age (C57BL/6-Jrj and Swiss, Janvier Labs France) and kept until full sexual maturity. For all experiments, mice were euthanized by cervical dislocation between 9 and 16 weeks of age by trained personnel. Mice have not been involved in any procedure except genotyping prior to euthanasia.

## Generation of mouse strains, husbandry, and genotyping

*separase^LoxP/LoxP*, *separase^LoxP/LoxP Zp3 Cre^+* (Kudo et al, 2006), *Pttg^−/−* (Wang et al, 2001), and *separase^LoxP/LoxP Zp3 Cre^+ Pttg^−/−* mice were house-bred in the animal facilities of UMR7622 (*separase^LoxP/LoxP Zp3 Cre^+* and *Pttg^−/−* strain) and UMR7592 (mice of all three strains used for experiments in this study). Primer sequences used

for genotyping are available upon request. Littermates were used as controls in experiments. Replicate experiments were performed with mice from different litters.

## Mouse oocyte harvesting and in vitro culture

Fully grown prophase I-arrested oocytes were harvested from dissected ovaries. Oocytes were retrieved by mouth pipetting follicles through narrow hand-pulled glass pasteur pipets. Selected fully grown prophase I-arrested oocytes were cultured at 38 °C in drops of self-made M2 media supplemented with 100 μg/mL dbcAMP (dibutyryl cyclic AMP, Sigma-Aldrich, D0260) and covered with mineral oil (suitable for embryo culture, FUJIFILM Irvine Scientific, 9305). To induce resumption of meiosis I from prophase I, and meiotic maturation, oocytes were washed three times in abundant drops of M2 media. Oocytes were synchronized visually at GVBD (Germinal vesicle breakdown), and only oocytes undergoing GVBD within 90 min after release were used (El Jailani et al, 2024). For meiosis II experiments, oocytes were always harvested in M2 media and then cultured overnight in M16 media (homemade). To study Mad2 recruitment to kinetochores as a readout of APC/C activity, oocytes were treated with the spindle-depolymerizing drug nocodazole (Sigma-Aldrich, M1404). Oocytes were cultured in M2 media supplemented with nocodazole from GVBD until chromosome spreads were performed, at a final concentration of 400 nM. For activation experiments in meiosis II, oocytes were cultured in M2 media after overnight incubation in M16. Strontium chloride (SrCl2, Sigma-Aldrich, 204463) was used to release oocytes from metaphase II arrest. At 16.5 h after GVBD, oocytes of the indicated genotypes were incubated in M2 medium without CaCl2 (activation medium, homemade) for 1 h, and transferred into activation medium supplemented with 100 mM SrCl2 (Bouftas et al, 2022). After washing the oocytes in 6 droplets of activation medium, they were observed by live imaging microscopy to visualize release from CSF arrest.

## In vitro transcription and oocyte microinjection

Plasmids used are described in the Reagents and Tools Table. The cleavage sensor H2B-mCherry-Scc1-YFP construct (Nikalayevich et al, 2018) was cloned into a pRN3-T3 plasmid. Plasmids to express separase and securin-YFP have been described (Kudo et al, 2006; Touati et al, 2012). The human cyclin B1-GFP plasmid (described previously in (Herbert et al, 2003)) was used as a template to generate a kinase-dead hydrophobic patch mutant (MRL mutant), using the Q5 site-directed mutagenesis kit to mutate the MRAIL motif to AAAIA. Primer sequences used for PCR cloning are available upon request. Plasmids were digested and linearized with the restriction enzyme Sfi1. Corresponding mRNAs were obtained by performing in vitro transcription using the mMessage mMachine T3 Kit (Invitrogen, AM1348). After 3 h of transcription, mRNAs were purified by using the RNeasy Mini Kit 5 (Qiagen, 74104). Purified mRNAs were injected into GV- or metaphase II-arrested oocytes. GV- and metaphase II-arrested oocytes were left for 1 h to allow expression of protein before release from prophase I-arrest (GV-arrested oocytes) or start of the movie (with or without activation with strontium, metaphase II-arrested oocytes). Microinjection pipettes were made by pulling capillaries (Harvard Apparatus, 30-0038) with a magnetic puller

(Next Generation Micropipette Puller P-1000, Sutter Instrument). Oocytes were manipulated on an inverted Nikon Eclipse Ti microscope equipped with Eppendorf micromanipulators, with a holding pipette (Eppendorf, VacuTip I EP5195000036-25, and Vitrolife, 15331), and injections were done using a FemtoJet Microinjection pump (Eppendorf) with continuous flow settings.

## Chromosome spreads

Tyrode's acid treatment (El Jailani et al, 2024) was used to dissolve the *zona pellucida* (ZP) of oocytes. ZP-free oocytes were fixed in hypotonic solution containing 1% paraformaldehyde (Sigma-Aldrich 441244), 0,15% Triton X-100 (Sigma-Aldrich, T8787) and 3 mM DTT (Sigma-Aldrich, D9779-1G) diluted in $H_2O$. Chromosome spreads were performed as described in (Bouftas et al, 2022), at the indicated time points, using a microscopic ten-well slide (Fischer Scientific, 10588681).

## Immunostaining

To stain chromosome spreads, a first step of saturation was done with PBS supplemented with 3% BSA (Sigma-Aldrich, A4378), final concentration. Spreads were incubated at room temperature for 2 h with primary antibodies, washed with PBS and incubated 1 h with secondary antibodies. Centromeres were stained with human ACA primary antibody (anti-centromere protein antibody, Cliniciences, 15-234, 1:100). To stain for proteins involved in centromeric cohesion protection, we used rabbit polyclonal anti-Sgo2 generated against the last 20 aa in the C-terminus (self-made, 1:100), and mouse monoclonal anti-PP2AC-subunit clone 1D6 conjugated with Alexa Fluor 488 (Sigma-Aldrich, 05-421-AF488, 1:100). To stain Mad2, we used rabbit polyclonal anti-Mad2 (self-made, 1:150) (Wassmann and Benezra, 1998). The following secondary antibodies were used at 1:200: donkey anti-human Cy3 secondary antibody (Jackson Immuno Research, 709-166-149), donkey anti-human Alexa Fluor 647 (Jackson Immuno Research, 709-606-149), donkey anti-rabbit Cy3 (Jackson Immuno Research, 711-166-152), donkey anti-rabbit Alexa Fluor (Jackson Immuno Research, 711-546-152), donkey anti-mouse Alexa Fluor 488 (Jackson Immuno Research, 715-546-150). Rec8 staining was done as described (Gryaznova et al, 2021). Slides were mounted with Vectashield mounting medium supplemented with DAPI (EUROBIO Scientific, H-1200). Acquisitions of stained chromosome spreads were made with either a Zeiss Axiovert 200 M inverted microscope or a Nikon Eclipse Ti2-E inverted microscope, both coupled with an EMCD camera (Evolve 512, Photometrics), a Yokogawa CSU-X1 spinning disk, and equipped with appropriate lasers and filter-sets for three-color imaging, using a 100×/1.4 NA oil Dic Plan-Apochromat Zeiss objective or, 100×/1.45 oil Plan-Apochromat Nikon objective. 6 z-sections with 0.4 mm interval were taken. Images were acquired by Metamorph software (Metamorph 7.10.3) and mounted with Fiji software (ImageJ2 2.9.0). No manipulations were made other than brightness and contrast adjustments that were applied with the same settings to images being compared.

## Live imaging

Live imaging of the cleavage sensor in mouse oocytes was performed under temperature-controlled conditions with a motorized inverted Nikon Eclipse Ti2-E microscope equipped with an EMCD camera (Evolve 512, Photometrics) or a high-sensitivity sCMOS camera (Photometrics), and a PrecisExite High Power LED Fluorescence, using a 20×/0.75 NA oil Plan-Apochromat Nikon objective. Chambers with oocytes microinjected with mRNAs where indicated, were prepared for imaging as described in Nikalayevich et al (2018). Time lapse was set up for 16 h in meiosis I, with time points taken every 20 mins. 15–20 z-sections with 3 μm distance were acquired in mCherry and YFP channels for the cleavage sensor, and 1-z section for the TRANS image. For live imaging of mouse oocytes in meiosis II, a motorized inverted Nikon Eclipse Ti2-E microscope equipped with a Yokogawa CSU-X1 spinning disk, coupled with an EMCD camera (Evolve 512, Photometrics), with appropriate lasers and filter-sets for three-color imaging and a 20×/0.75 NA oil Plan-Apochromat Nikon objective was used. Time lapse was set up for 6 h with 10 min intervals. Prior to acquisition, metaphase II oocytes were injected with mRNAs as indicated, preincubated in M2 media supplemented with 1 mM SiR-DNA (far-red DNA labeling probe, Spirochrome, SC007) for 1 h. 15–20 z-sections with 3 μm distance were acquired in the 640 nm channel for SiR-DNA, and 1-z section for the TRANS image. All time-lapse acquisitions were piloted by Metamorph software (Metamorph 7.10.3). Stills of movies were mounted with Fiji software (ImageJ2 2.9.0) and contrast/brightness were adjusted equally to all conditions being compared.

## Quantification and statistical analysis

Quantifications were made using Fiji software. To quantify Sgo2, PP2A, and Mad2 total signal on chromosome spreads, levels of fluorescence were measured on sum-projected images. A square of $10 \times 10$ pixels was drawn on centromeres as depicted on the scheme in Fig. 3G,I. The signals measured (Ifluo) were corrected to background (bg) and normalized to ACA signals as follows: Ifluo(normalized) = [Ifluo − Ifluobg]/[IfluoACA − IfluoACA(bg)]. To quantify the mutant cyclin B1(MRL)-GFP degradation, mean levels of fluorescence intensities were measured on sum-projected time-lapse images. A square of $100 \times 100$ pixels was drawn in the oocyte. Mean levels of fluorescence were measured for each timepoint of the time-lapse movie and were corrected to background and normalized to the value of the first timepoint (t0). All data plots and statistical analysis were obtained with PRISM 10 software. To monitor separase activation timing, anaphase I onset was used as our readout. Error bars with standard deviation (SD), and mean bars are represented. Statistical analysis was performed using nonparametric Mann–Whitney $U$ test (ns: not significant, $**P < 0.01$, $****P < 0.0001$). All experiments were done at least three times independently. Shown are results from all replicates performed or, of one representative replicate, where indicated.

# Data availability

This study includes no data deposited in external repositories.

The source data of this paper are collected in the following database record: biostudies:S-SCDT-10_1038-S44318-025-00522-0.

## Peer review information

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

## Acknowledgements

We thank lab members for discussions and useful suggestions, Warif El Yakoubi for preliminary experiments, and Esther Chanty for help with statistical tests. We thank Alberto Pendas (University of Salamanca) for providing *Pttg*$^{-/-}$ sperm straws, and Nicolas Minc (Institut Jacques Monod) for access to a Sutter Instruments Micropipette puller. We are grateful to Isabelle Le Parco and members of the animal house "Animalerie Buffon", as well as the informatics and administrative support services of the Institut Jacques Monod. SEJ received a 4th-year PhD fellowship from the Fondation pour la Recherche Médicale (Fin de thèse, FDT202404018087). Financial support for this work was obtained by the Fondation pour la Recherche Médicale (Equipe FRM DEQ 202103012574), and the Agence Nationale de la Recherche (ANR-23-CE13-0015-01) to KW.

## Author contributions

**Safia El Jailani**: Data curation; Formal analysis; Investigation; Visualization; Methodology; Writing—original draft. **Damien Cladière**: Resources; Data

curation; Formal analysis. **Elvira Nikalayevich**: Data curation; Formal analysis. **Sandra A Touati**: Data curation; Formal analysis; Investigation. **Vera Chesnokova**: Investigation. **Shlomo Melmed**: Resources. **Eulalie Buffin**: Data curation; Supervision; Investigation. **Katja Wassmann**: Conceptualization; Supervision; Funding acquisition; Investigation; Writing—original draft; Project administration; Writing—review and editing.

Source data underlying figure panels in this paper may have individual authorship assigned. Where available, figure panel/source data authorship is listed in the following database record: biostudies:S-SCDT-10_1038-S44318-025-00522-0.

## Disclosure and competing interests statement

The authors declare no competing interests.

# Expanded View Figures

**Figure EV1.   Related to Fig. 1B.**

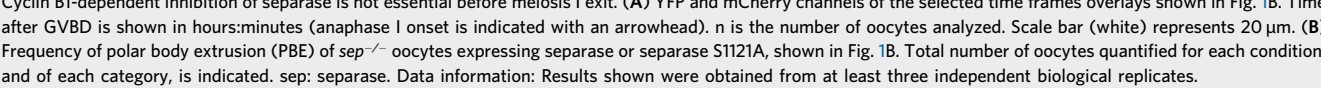

Cyclin B1-dependent inhibition of separase is not essential before meiosis I exit. (**A**) YFP and mCherry channels of the selected time frames overlays shown in Fig. 1B. Time after GVBD is shown in hours:minutes (anaphase I onset is indicated with an arrowhead). n is the number of oocytes analyzed. Scale bar (white) represents 20 μm. (**B**) Frequency of polar body extrusion (PBE) of $sep^{-/-}$ oocytes expressing separase or separase S1121A, shown in Fig. 1B. Total number of oocytes quantified for each condition, and of each category, is indicated. sep: separase. Data information: Results shown were obtained from at least three independent biological replicates.

**A**

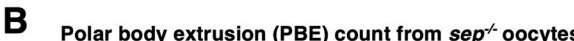

control (n=36)    control (n=23)

7:30  7:50  8:10  8:30  8:50

mCherry

YFP

mCherry/YFP

*sep*⁺/⁺

+ separase (n=110)    + separase (n=70)

+ separase S1121A (n=58)    + separase S1121A (n=56)

*sep*⁻/⁻

**B**

Polar body extrusion (PBE) count from *sep*⁻/⁻ oocytes

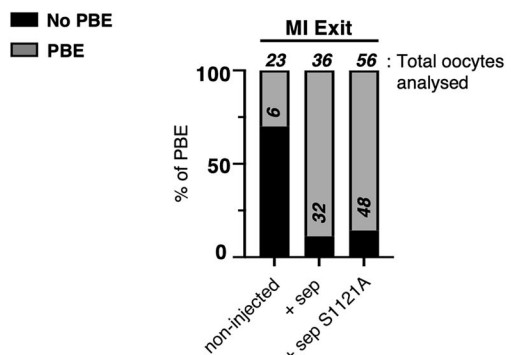

■ No PBE
■ PBE

MI Exit

23   36   56   : Total oocytes analysed

% of PBE

100

50

non-injected    + sep    + sep S1121A

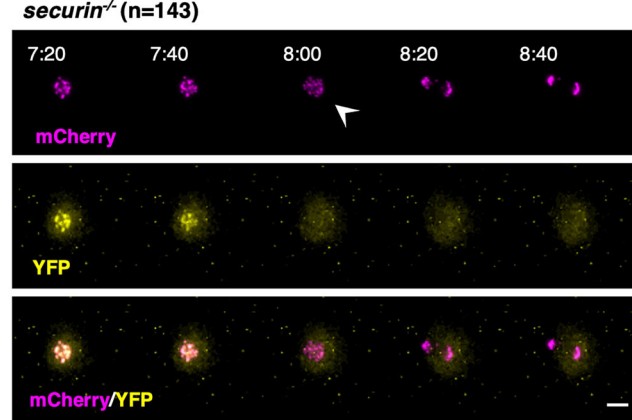

**Figure EV2.  Related to Fig. 2A.**

Securin is not essential for separase inhibition in meiosis I. YFP and mCherry channels of the selected time frames overlays shown in Fig. 2A. Time after GVBD is shown in hours:minutes (anaphase I onset is indicated with an arrowhead). *n* is the number of oocytes analyzed. Scale bar (white) represents 20 µm. Data information: Results shown were obtained from at least three independent biological replicates.

## A

**control (n=15)**

| GVBD | 1:20 | 2:40 | 4:0 | 5:20 | | 7:40 | 8:20 | 9:00 | 9:40 | 10:20 |

mCherry

YFP

mCherry/YFP

**+ separase (n=29)**

| GVBD | 1:20 | 2:40 | 4:0 | 5:20 | | 7:40 | 8:20 | 9:00 | 9:40 | 10:20 |

**+ separase S1121A (n=65)**

| -00:20 | GVBD | 1:20 | 2:40 | 4:00 | | 7:40 | 8:20 | 9:00 | 9:40 | 10:20 |

*sep⁻/⁻ securin⁻/⁻*

## B

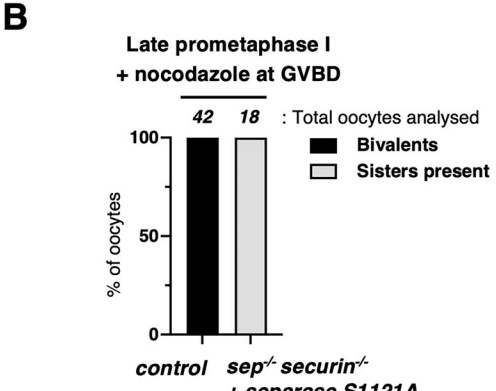

**Late prometaphase I
+ nocodazole at GVBD**

42    18    : Total oocytes analysed

■ Bivalents
□ Sisters present

% of oocytes

100

50

*control*    *sep⁻/⁻ securin⁻/⁻
+ separase S1121A*

◀

**Figure EV3.   Related to Fig. 3A,C; Movie EV2 and Appendix Fig. S1.**

Complete loss of separase control reveals absence of cohesin protection in early prometaphase I. (**A**) YFP and mCherry channels of selected time frames overlays shown in Fig. 3A. Time after GVBD is shown in hours:minutes (cleavage onset is indicated with an arrowhead). *n* is the number of oocytes analyzed. Scale bar (white) represents 20 μm. Complete movies are shown in Appendix Fig. S1. (**B**) Frequency of chromosome categories observed at late prometaphase I (6 h after GVBD) quantified from chromosome spreads of wild-type and *sep*$^{-/-}$*securin*$^{-/-}$ oocytes expressing separase S1121A shown in Fig. 3C. Oocytes were treated with nocodazole at GVBD. Total number of chromosomes quantified for each condition is indicated. sep: separase. Data information: Results shown were obtained from at least three independent biological replicates.

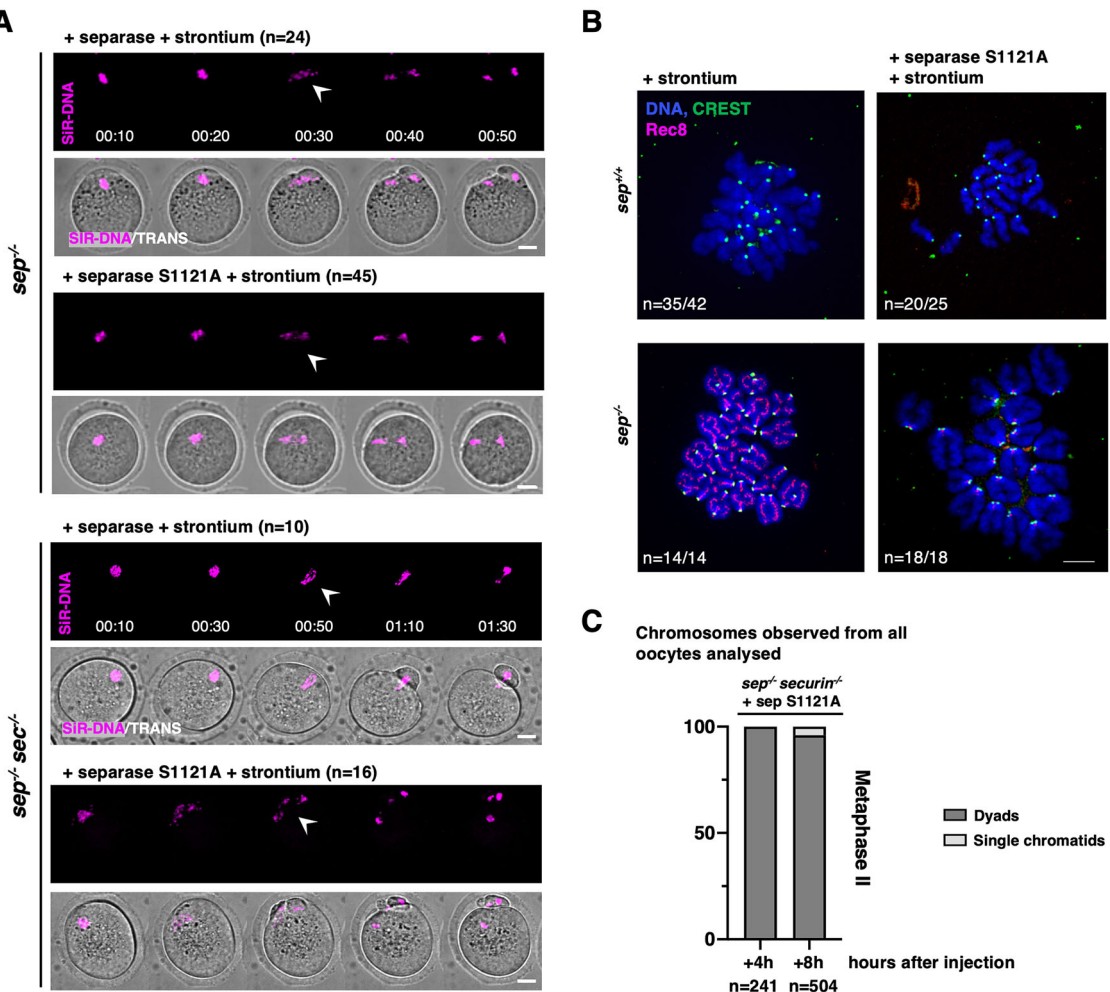

**Figure EV4. Related to Figs. 4B,D and 5B,D.**

When rescued in metaphase II, *sep⁻/⁻* and *sep⁻/⁻securin⁻/⁻* oocytes can be activated to undergo meiosis II and separate chromosomes. (A) Live imaging movies of *sep⁺/⁺* (top) and *sep⁻/⁻ securin⁻/⁻* (bottom) oocytes in Figs. 4B and 5B. Where indicated, oocytes were chemically induced with strontium to verify sufficient expression of wild-type separase or separase S1121A. For all conditions, oocytes were injected with the constructs around 16 h after GVBD, and were subjected to live imaging around 18 h after GVBD, corresponding to 5 min after oocytes were activated with strontium. Time after start of the movie is shown in hours:minutes (anaphase II onset is indicated with an arrowhead). n is the number of oocytes analyzed. Scale bar (white) represents 20 μm. (B) *sep⁺/⁺* (top) or *sep⁻/⁻* (bottom) oocytes were chemically induced with strontium and fixed for chromosome spreads around 20 h after GVBD (Metaphase II). Where indicated, oocytes were injected with the separase S1121A construct around 16 h after GVBD, and were fixed around 20 h after GVBD, corresponding to 45 min after oocytes were activated with strontium. Centromeres/kinetochores were stained with CREST (green), cohesin with anti-Rec8 antibody (red) and chromosomes with DAPI (blue). A representative spread and magnification of one chromosome (white dashed line squares, insert at the bottom right corner) are shown for each condition. n indicated the total number of chromosomes analyzed. Scale bars (white) represents 10 μm. (Related to Figs. 4D and 5D). (C) Frequency of chromosome categories observed at metaphase II quantified from *sep⁻/⁻securin⁻/⁻* oocytes expressing separase S1121A. Oocytes were injected with the construct around 16 h after GVBD and were fixed for chromosome spreads at the indicated timings (4 h or 8 h after injections). Total number of chromosomes quantified for each condition, is indicated. sep: separase. (Related to Fig. 5D). Data information: Results shown were obtained from at least three independent biological replicates.

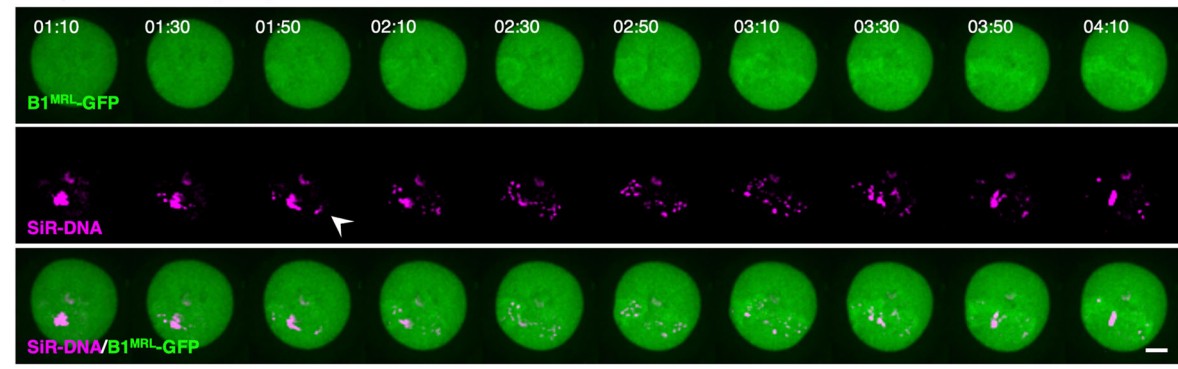

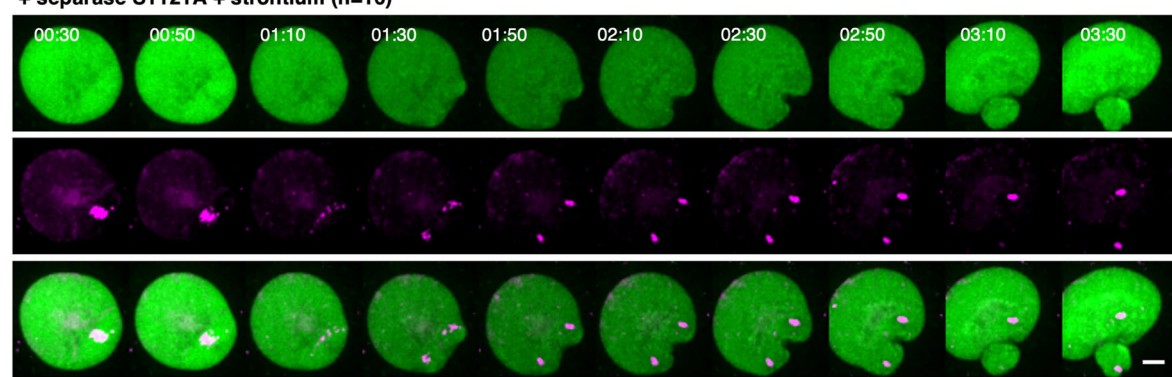

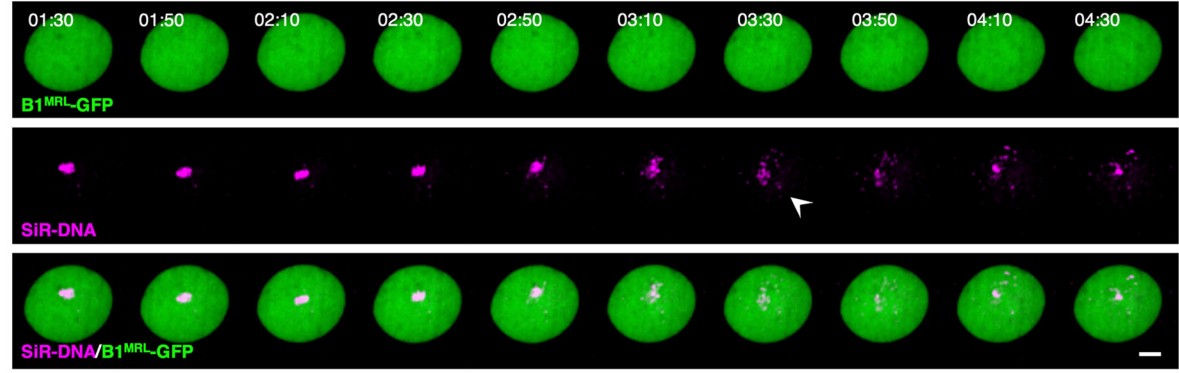

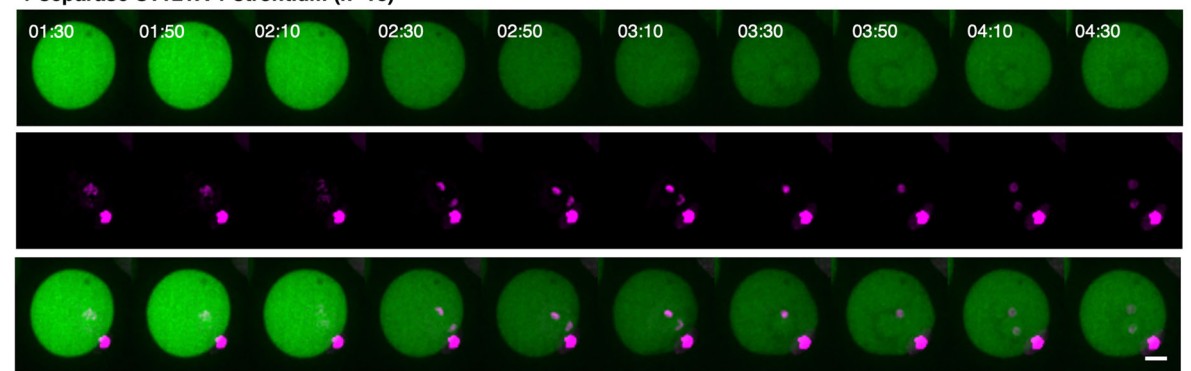

◀ **Figure EV5.   Related to Fig. 6D,F.**

Separase-out-of-control activation is independent of APC/C activation in meiosis II. Overlays of the GFP and Far-Red channels of the selected time frames shown in Fig. 6D,F. $sep^{-/-}$ $securin^{-/-}$ (top) and $securin^{-/-}$ (bottom) oocytes were injected with separase S1121A and the GFP-cyclin B1 MRL mutant around 16 h after GVBD, and were subjected to live imaging around 18 h after GVBD. Where indicated, oocytes were activated with strontium around 5 min before start of live imaging. Time after start of the movie is shown in hours:minutes (separation of chromosomes is indicated with an arrowhead). n is the number of oocytes analyzed. Scale bar (white) represents 20 μm. Data information: Results shown were obtained from at least three independent biological replicates.

