## [Peer Review File · The EMBO Journal]

Eliminating separase inhibition reveals absence of robust cohesin protection in oocyte metaphase II

Safia El Jailani, Damien Cladière, Elvira Nikalayeich, Sandra Touati, Vera Chesnokova, Shlomo Melmed, Eualie Buffin, and Katja Wassmann

Corresponding author(s): Katja Wassmann (katja.wassmann@ijm.fr)

Review Timeline:

Submission Date:	10th Feb 25
Editorial Decision:	11th Mar 25
Revision Received:	9th Jun 25
Editorial Decision:	14th Jul 25
Revision Received:	15th Jul 25
Accepted:	18th Jul 25

Editor: Hartmut Vodermaier

Transaction Report:

Dr. Katja Wassmann
CNRS - Institut Jacques Monod
15 rue Helene Brion
PARIS CEDEX 13, Paris 75013
France

11th Mar 2025

Re: EMBOJ-2025-120451
Elimination of separase inhibition reveals absence of cohesin protection in oocyte metaphase II

Dear Katja,

Thank you for submitting your study on separase and cohesin protection control during meiosis in mouse oocytes for our consideration. It has now been reviewed by three expert referees, whose comments are copied below. As you will see, all referees appreciate that importance of the present questions and the high technical quality of your experimental system for studying them in live oocytes. However, while referees 1 and 2 only have a couple of specific suggestions and queries, referee 3 raises several well-taken concerns regarding the claimed lack of centromeric cohesion protection at particular meiotic stages.

In light of these comments, I feel that this work for could become suitable for EMBO Journal publication, but would require adequate clarification of these key issues, either through additional data or appropriate toning down/qualification of certain conclusions. In order to discuss how this might best be achieved, I would encourage you to contact me with a preliminary point-by-point response already during the early stages of your revision work. Please be reminded that it is our policy to allow only a single round of major revision, making it important to comprehensively respond to all referee points at the time of resubmission.

Detailed information on preparing, formatting and uploading a revised manuscript can be found below and in our Guide to Authors. Thank you again for the opportunity to consider this work for The EMBO Journal, and I look forward to hearing from you in due time.

With kind regards,

Hartmut

4) Each main and each Expanded View (EV) figure should be uploaded as individual production-quality files (preferably in .eps, .tif, .jpg formats). For suggestions on figure preparation/layout, please refer to our Figure Preparation Guidelines:

- 5) Point-by-point response letters should include the original referee comments in full together with your detailed responses to them (and to specific editor requests if applicable), and also be uploaded as editable (e.g., .docx) text files.
- 6) Please complete our Author Checklist, and make sure that information entered into the checklist is also reflected in the manuscript; the checklist will be available to readers as part of the Review Process File. A download link is found at the top of our Guide to Authors: embopress.org/page/journal/14602075/authorguide
- 7) All authors listed as (co-)corresponding need to deposit, in their respective author profiles in our submission system, a unique ORCID identifier linked to their name. Please see our Guide to Authors for detailed instructions.
- 8) Please note that supplementary information at EMBO Press has been superseded by the 'Expanded View' for inclusion of additional figures, tables, movies or datasets; with up to five EV Figures being typeset and directly accessible in the HTML version of the article. For details and guidance, please refer to: embopress.org/page/journal/14602075/authorguide#expandedview
- 9) To facilitate reproducibility and cross-laboratory adoption of methodologies, please structure the Materials & Methods section as outlined in our guide to authors, including a completed Reagents and Tools Table that can be downloaded from our author guidelines as well (<https://www.embopress.org/page/journal/14602075/authorguide#structuredmethods>).
- 10) Digital image enhancement is acceptable practice, as long as it accurately represents the original data and conforms to community standards. If a figure has been subjected to significant electronic manipulation, this must be clearly noted in the figure legend and/or the 'Materials and Methods' section. The editors reserve the right to request original versions of figures and the original images that were used to assemble the figure. Finally, we generally encourage uploading of numerical as well as gel/blot image source data; for details see: embopress.org/page/journal/14602075/authorguide#sourcedata

At EMBO Press, we ask authors to provide source data for the main manuscript figures. Our source data coordinator will contact you to discuss which figure panels we would need source data for and will also provide you with helpful tips on how to upload and organize the files.

In the interest of ensuring the conceptual advance provided by the work, we recommend submitting a revision within 3 months (9th Jun 2025). Please discuss the revision progress ahead of this time with the editor if you require more time to complete the revisions. Use the link below to submit your revision:

Link Not Available

Referee #1:

The manuscript "Elimination of separase inhibition reveals absence of cohesin protection in oocyte metaphase II" by El Jailani et al., focuses on mechanisms controlling the activity of Separase and protection of cohesion during mouse meiosis I and II. As a main tool, authors used combination of genetic deletion of Securin and Separase and their combination, together with microinjection of Separase resistant to phosphorylation by CDK1 (Separase S1121A).

Authors confirmed previous findings, that inhibition of Separase, either by Securin or by CDK1 phosphorylation, is sufficient for control of Separase activity in meiosis I. However, they newly show that even in the presence of Securin, the replacement of wild type Separase by Separase mutation S1121A led into precocious separation of some sister chromatids after prolonged arrest in metaphase II.

They show, that the simultaneous removal of Securin and CDK1 control of Separase in GV oocytes led into loss of Separase control, with cohesin cleavage scheduled imminently after GVBD. The simultaneous conversion of bivalents into single chromatids indicates, that during GVBD, cohesion is unprotected.

They also show, that in meiosis II oocytes, simultaneous loss of Securin and CDK1 control of Separase leads into cohesin cleavage, and this does not require APC/C activity.

Major concerns:

1. Is it possible to rescue early cohesin cleavage in Separase and Securin depleted oocytes overexpressing Separase S1121A by coinjection of Securin? I believe that this is an important control, which is missing.
2. From the Figure 3A and movie EV2 it seems that chromosomes after cleavage by precociously activated Separase, which turned them into single chromatids, are not showing movements along the spindle axis. Could authors comment on that?
3. It was shown independently before (Nabti et al, 2008 and Chiang et al., 2011) that sole removal of Securin by morpholino led into separation of sisters in meiosis II. However, it seems that the genetic deletion of Securin has no such consequences. Could authors comment on that?

Minor concerns:

Authors should indicate when oocytes were harvested for spreads for MI exit group. (Figure 1G and F).

Overall, this manuscript brings important information, concerning not so well understood mechanisms in oocyte meiosis, namely control of Separase activity and regulation of protection of cohesion, both of which might have impact on overall high incidence of aneuploidy in these cells. Therefore, I suggest this manuscript for publication in EMBO journal, after resolving the above concerns.

Referee #2:

Review of EMBO Journal manuscript 2025-120451 entitled 'Elimination of separase inhibition reveals absence of cohesin protection in oocyte metaphase II' by Safia El Jailani, Damien Cladière, Elvira Nikalayevich, Sandra A Touati, Vera Chesnokova, Shlomo Melmed, Eualie Buffin, and Dr. Katja Wassmann

Meiosis is hallmarked by two rounds of chromosome segregations without an intermediate replication. Curiously, both anaphases are triggered by one and the same protease, separase. When homologous chromosomes are separated by separase-dependent cleavage of Rec8-cohesin distal from chiasmata in MI, sister chromatid cohesion is maintained by Sgol2-PP2A-dependent protection of pericentromeric Rec8. While it is clear that this last pool of cohesin must be rendered susceptible to separase prior to the separation of sisters in MII, many questions about the choreography of the meiotic chromosome dance remain to be answered. How is separase held inactive prior to anaphase I and II - by securin, CCC (cyclin B1-Ck1-Ck1/2) and/or Sgol2-Mad2? And when exactly is the protection of pericentromeric cohesion first established and then nullified?

Using oocytes of securin and/or separase knock-out mice in combination with mRNA-injection based expression of wild type or CCC-resistant separase variants at different stages of meiosis, El Jailani et al. report here the following key findings:

- i) Separase is redundantly controlled by both securin and CCC in both meiotic metaphases.
- ii) CCC, but not securin, keeps separase in check after anaphase I until metaphase II.
- iii) Murine Sgol2-Mad2 likely plays no or only a minor role in separase regulation during female meiosis.
- iv) Pericentromeric cohesion protection is established surprisingly late during early MI.
- v) Following a normal MI, pericentromeric cohesion is no longer protected in metaphase II.
- vi) Separase-dependent individualization of sister kinetochores in anaphase I is pre-condition for deprotection of pericentromeric cohesion.

This is a very impressive body of work. The experiments are all very well controlled, and the beautiful data fully support the conclusions. I enthusiastically recommend publication in the EMBO Journal and have only one request (which does not involve any additional experiment), one commentary (which, I feel, touches an important issue, but is nevertheless only a remark) and a couple of minor questions/suggestions:

1) A common denominator of two previous studies was that separase has a protease-independent function in cytokinesis at the end of MI (Kudo et al., 2006; Gorr et al., 2006). However, there was a seeming contradiction whether or not this required the Cdk1-inhibitory role of separase. El Jailani et al. have the data to clarify this issue. Therefore, they should show in an additional figure the PB1 extrusion rates of sep^{-/-} oocytes that were micro-injected to express WT- versus S1126A-separase.

2) Surprisingly, in the discussion the authors compare their results with those of Shindo et al., 2022. The model put forward in that paper is the following: Sole inhibition of separase by securin in metaphase » SAC satisfaction and APC/C activation » liberation of separase, which then cleaves itself without touching cohesin » re-inhibition of separase by CCC » more APC/C activity and more cyclin B1 degradation » 2nd liberation of separase which now finally has a taste for cohesin. I think that everyone in the field, except for maybe the authors, would agree that this cannot be true. This model also does not fit what Wassmann and co-workers observe in MI. What they see is: Redundant inhibition of separase by securin and CCC in metaphase of MI » APC/C activation » separation of bivalents into dyads » re-inhibition of separase by residual CCC during the transition into MII. This is, however, exactly what Stemmann and co-workers observed for mitosis (Hellmuth et al., 2014, 2015). In trying to explain the re-formation of separase-CCC, El Jailani et al. then continue to speculate that S1126 phosphorylation is removed by separase-associated PP2A. While this is an obvious model, there is no evidence for it. (In fact, we have tried very hard to prove this but eventually had to apprehend that this is not what is happening.) In contrast, there is published evidence that dephosphorylation of cyclin B1 at the end of mitosis vastly increases the affinity of residual CCC towards separase (Hellmuth et al., 2015).

Minor points:

A third inhibitor of separase, namely Sgo2-Mad2, is present in somatic cells arrested in mitosis due to SAC activation (Hellmuth et al., 2020). This sentence may make the impression to the reader that separase inhibition by Sgo2-Mad2 in human somatic cells is limited to mitosis. But it is not....

Page 5: 'It has been proposed that pericentromeric cohesin protection is maintained until metaphase II to prevent separase from cleaving this fraction of cohesin holding sister chromatids together, as separase may become reactivated upon degradation of cyclin B1 and securin at meiosis I exit.'

Can the authors cite the corresponding study?

The authors observe that Mad2 staining was reduced in metaphase I even when bivalents were prematurely separated into sisters due to expression of separase-S1126A in separase^{-/-} securin^{-/-} oocytes. However, Mad2 staining could be rescued by nocodazole treatment. Does that mean that the SAC in meiosis I oocytes senses only attachment and no tension?

Page 10: 'Using sep^{-/-} oocytes, we set out to determine whether cyclin B1 alone is indeed not required for separase inhibition in meiosis II.'

This sentence would be easier to understand without the 'alone'.

As Cdk1 inactivation is required for anaphase movement, I recommend replacing segregation by separation in the subheading 'APC/C activation is not required for sister chromatid segregation in meiosis II'.

Typo on page 13: 'constitutive active'.

Page 14: 'However, to address the role of Sgo2-Mad2 convincingly, it will be necessary to invalidate Sgo2 or Mad2 specifically in metaphase I and not before, and find ways to separate a potential role in separase inhibition from other essential functions both proteins play in oocyte meiosis, such as cohesin protection, chromosome alignment and SAC control.'

At least for human Sgo2 and separase, such mutants have been identified (Hellmuth et al., 2020).

Referee #3:

This manuscript investigates the regulation of separase activity during meiosis in oocytes, focusing on two known separase inhibitors, cyclin B1 and securin.

Firstly, the authors used a separase knockout mouse model and expressed a mutant form of separase (S1121A), which is not inhibited by cyclin B1, in oocytes. Although oocytes expressing separase S1121A normally segregated chromosomes during meiosis I, they resulted in precocious separation of sister chromatids by metaphase II. This indicates that cyclin B1-mediated separase inhibition is not necessary for meiosis I but is essential for maintaining centromeric cohesion until metaphase II. Based on this observation, the authors claim that centromeric cohesion protection is absent during the transition period from meiosis I to metaphase II. However, I am concerned about this claim, as I will discuss later (Major comment 1).

Secondly, they used a securin knockout mouse model and showed that oocytes lacking securin exhibited both meiosis I and meiosis II almost normally. These observations demonstrate that securin is not essential for meiosis, which is consistent with previous studies.

Thirdly, the authors tested oocytes lacking both cyclin B1-mediated and securin-mediated separase inhibitions by combining separase and securin knockouts. These oocytes showed precocious chromosome separation from bivalents to dyads or single chromatids during early prometaphase I. Based on this, the authors claim that centromeric cohesion protection is absent during early prometaphase, but I cannot be convinced of the appropriateness of this claim (Major comment 2).

Fourthly, the authors investigated how separase is inhibited during metaphase II. They used an approach where separase S1121A was introduced after culturing separase knockout oocytes to metaphase II. Separase S1121A did not induce chromosome separation, likely because it was inhibited by securin. Consistently, when both separase and securin were knocked out and separase S1121A was introduced at metaphase II, chromosome separation was induced. Although separation from bivalents to dyads was observed, separation to single chromatids did not occur, suggesting that centromeric cohesion protection was extended to metaphase II. The authors claim that this extension is due to the absence of kinetochore individualization during the transition from meiosis I to meiosis II, but I could not find evidence for this (Major comment 3).

Fifthly, the authors introduced separase S1121A in securin knockout oocytes after culturing to metaphase II. These oocytes exhibited precocious separation of dyad chromosomes to single chromatids. Based on this observation, the authors claim that centromeric cohesion is absent during metaphase II. However, once again, I am not convinced of the appropriateness of this claim (Major comment 4).

Overall, this paper uses a very clean experimental system taking advantage of separase and securin knockout oocytes to clearly demonstrate the regulation of separase inhibition by cyclin B1 and securin throughout meiosis. While neither cyclin B1 nor securin is essential for inhibiting separase on their own, their combined disruption leads to separase activation, a phenotype expected from previous reports. The authors base their conclusions about the presence or absence of centromeric cohesion protection at various stages of meiosis on the patterns of chromosome separation caused by ectopic separase activation, which is reflected in the title and overall claim throughout the paper. However, as discussed below, I have reservations about their

conclusions on the absence of centromeric cohesion protection at several stages of meiosis. I appreciate the beauty of this paper as a clear and robust demonstration of separase inhibition by double inhibitors cyclin B1 and securin in mouse oocytes, which is a great contribution to the field. However, I feel that the paper would benefit from removing or toning down their conclusion about the absence of centromeric cohesion protection at some meiotic stages.

Major comment:

1- The authors claim that centromeric cohesion protection is absent during the transition from meiosis I to meiosis II, based on their observation that separase S1121A causes precocious chromosome separation by metaphase II. However, I would interpret their observation to mean that centromeric cohesion protection does exist but centromeric cohesion is gradually lost due to prolonged separase activity, leading to precocious chromosome separation. The difference in interpretation between the authors and myself might stem from a difference in assumptions about centromeric cohesion protection. The authors might assume that if centromeric cohesion is protected, there should be no reduction in cohesin at the centromeres at all. However, this assumption is less likely to be correct. A more general assumption is that if centromeric cohesion is protected, there is less reduction in cohesin at centromeres compared to chromosome arms. If this more likely assumption is considered, it is natural that centromeric cohesion would eventually be lost if separase activity is prolonged by S1121A even in the presence of centromeric cohesion protection.

2- The authors claim that centromeric cohesion protection is absent during early prometaphase I, based on their observation that chromosome bivalents are precociously separated into dyads or single chromatids during early prometaphase I when both cyclin B1-mediated and securin-mediated inhibitions are simultaneously disrupted. For the same reasons mentioned above, I cannot be convinced that this claim is appropriate. Indeed, their data show that there is a fraction of chromosomes that resulted in dyads, consistent with the idea that chromosomes separated from bivalents to dyads, and subsequently from dyads to single chromatids. This sequential chromosome separation would support the presence of centromeric cohesion protection.

3- In the experiments shown in Figure 5, the authors demonstrate that when separase activation occurs for the first time at metaphase II, separation from bivalents to dyads occurs, but not to single chromatids. This suggests that centromeric cohesion protection is extended to metaphase II in this experimental condition. The authors claim that this extension is due to the absence of separase-dependent kinetochore individualization during the transition from meiosis I to metaphase II. However, I cannot find evidence to support this. What the actual data show is that passing through meiosis I without separase activation leads to the extension of centromeric cohesion protection to metaphase II. That is, separase activity is needed to cancel centromeric cohesion protection during the meiosis I-II transition. To support their claim that this cancellation is mediated by separase-dependent kinetochore individualization, the authors should show that forced kinetochore individualization in the absence of separase activity results in a cancellation of centromeric cohesion protection.

4- In the experiments shown in Figure 6, the authors show that a second round separase activation at metaphase II results in precocious separation of chromosome dyads into single chromatids. Based on this observation, the authors claim that centromeric cohesion is absent during metaphase II. However, for the same reasons mentioned in comment 1, I am afraid that this claim is not appropriate. If ectopic separase activity is present, it is natural that even if centromeric cohesion protection exists, it would eventually lead to the separation into single chromatids.

5- One of the main targets of this study is mechanisms of centromeric cohesion protection during the transition from meiosis I to meiosis II in mouse oocytes. One such mechanism known during this period is the centromeric SUMO pathway (PMID: 29754905), yet the authors ignore this paper throughout their manuscript.

Minor comment:

It would be beneficial to add quantification of the separase activity sensor in Fig. 3A to strengthen the results.

Point-by point reply to the referee's comments:

We sincerely thank all three referees for having taken the time to evaluate our manuscript in depth, and for their comments and suggestions. Below please find the point-by-point reply to the issues raised. Changes in the manuscript in response to the reviewers are marked up in yellow throughout the manuscript.

Referee #1:

The manuscript "Elimination of separase inhibition reveals absence of cohesin protection in oocyte metaphase II" by El Jailani et al., focuses on mechanisms controlling the activity of Separase and protection of cohesion during mouse meiosis I and II. As a main tool, authors used combination of genetic deletion of Securin and Separase and their combination, together with microinjection of Separase resistant to phosphorylation by CDK1 (Separase S1121A).

Authors confirmed previous findings, that inhibition of Separase, either by Securin or by CDK1 phosphorylation, is sufficient for control of Separase activity in meiosis I. However, they newly show that even in the presence of Securin, the replacement of wild type Separase by Separase mutation S1121A led into precocious separation of some sister chromatids after prolonged arrest in metaphase II.

They show, that the simultaneous removal of Securin and CDK1 control of Separase in GV oocytes led into loss of Separase control, with cohesin cleavage scheduled imminently after GVBD. The simultaneous conversion of bivalents into single chromatids indicates, that during GVBD, cohesion is unprotected.

They also show, that in meiosis II oocytes, simultaneous loss of Securin and CDK1 control of Separase leads into cohesin cleavage, and this does not require APC/C activity.

We thank this reviewer for his/her comments and suggestions.

Major concerns:

1. Is it possible to rescue early cohesin cleavage in Separase and Securin depleted oocytes overexpressing Separase S1121A by coinjection of Securin? I believe that this is an important control, which is missing.

We thank the reviewer for pointing out that this control was missing, and have added the control experiment to Figure 3D and E.

2. From the Figure 3A and movie EV2 it seems that chromosomes after cleavage by

precociously activated Separase, which turned them into single chromatids, are not showing movements along the spindle axis. Could authors comment on that?

Under these conditions cohesin is removed so early that chromosomes and sister chromatids just fall apart, because at this time, no proper bipolar spindle has been formed yet. Indeed, spindle formation in mouse oocytes starts with a spherical belt transitioning to a prometaphase belt at the time when sisters are already separated in our experiment (see Kitajima et al, Cell 2011).

3. It was shown independently before (Nabti et al, 2008 and Chiang et al., 2011) that sole removal of Securin by morpholino led into separation of sisters in meiosis II. However, it seems that the genetic deletion of Securin has no such consequences. Could authors comment on that?

We can only speculate to why morpholino knock-down resulted in a phenotype that is not observed in the complete knock-out. In our hands, MO mediated knockdown leads to additional stress in oocyte culture, due to the long incubation times required for efficient knock-down, which may promote the appearance of certain phenotypes and explain the difference. This is why we had chosen to refrain from using transient approaches to study separase inhibition.

Minor concerns:

Authors should indicate when oocytes were harvested for spreads for MI exit group. (Figure 1G and F).

We state in the figure legend that oocytes were harvested immediately after polar body extrusion, around 8 hours after GVBD (Fig 1E, applies also to Figure 1F). This means that oocytes were visually selected and taken for spreads when a fully formed PB was visible, which took place on average 8 hours after GVBD (+/- 20 min).

Overall, this manuscript brings important information, concerning not so well understood mechanisms in oocyte meiosis, namely control of Separase activity and regulation of protection of cohesin, both of which might have impact on overall high incidence of aneuploidy in these cells. Therefore, I suggest this manuscript for publication in EMBO journal, after resolving the above concerns.

Referee #2:

Review of EMBO Journal manuscript 2025-120451 entitled 'Elimination of separase inhibition reveals absence of cohesin protection in oocyte metaphase II' by Safia El Jailani, Damien Cladière, Elvira Nikalayevich, Sandra A Touati, Vera Chesnokova, Shlomo Melmed, Eualie Buffin, and Dr. Katja Wassmann

Meiosis is hallmarked by two rounds of chromosome segregations without an intermediate replication. Curiously, both anaphases are triggered by one and the same protease, separase. When homologous chromosomes are separated by separase-dependent cleavage of Rec8-cohesin distal from chiasmata in MI, sister chromatid cohesion is maintained by Sgol2-PP2A-dependent protection of pericentromeric Rec8. While it is clear that this last pool of cohesin must be rendered susceptible to separase prior to the separation of sisters in MII, many questions about the choreography of the meiotic chromosome dance remain to be answered. How is separase held inactive prior to anaphase I and II - by securin, CCC (cyclin B1-Ck1-Ck1/2) and/or Sgol2-Mad2? And when exactly is the protection of pericentromeric cohesion first established and then nullified?

Using oocytes of securin and/or separase knock-out mice in combination with mRNA-injection based expression of wild type or CCC-resistant separase variants at different stages of meiosis, El Jailani et al. report here the following key findings:

- i) Separase is redundantly controlled by both securin and CCC in both meiotic metaphases.
- ii) CCC, but not securin, keeps separase in check after anaphase I until metaphase II.
- iii) Murine Sgol2-Mad2 likely plays no or only a minor role in separase regulation during female meiosis.
- iv) Pericentromeric cohesion protection is established surprisingly late during early MI.
- v) Following a normal MI, pericentromeric cohesion is no longer protected in metaphase II.
- vi) Separase-dependent individualization of sister kinetochores in anaphase I is pre-condition for deprotection of pericentromeric cohesion.

This is a very impressive body of work. The experiments are all very well controlled, and the beautiful data fully support the conclusions. I enthusiastically recommend publication in the EMBO Journal and have only one request (which does not involve any additional experiment), one commentary (which, I feel, touches an important issue, but is nevertheless only a remark) and a couple of minor questions/suggestions:

We highly appreciate the positive comments of this reviewer on our manuscript.

1) A common denominator of two previous studies was that separase has a protease-independent function in cytokinesis at the end of MI (Kudo et al., 2006; Gorr et al., 2006). However, there was a seeming contradiction whether or not this required the Cdk1-inhibitory role of separase. El Jailani et al. have the data to clarify this issue. Therefore, they should show in an additional figure the PB1 extrusion rates of sep^{-/-} oocytes that were micro-injected to express WT- versus S1126A-separase.

The reviewer is correct; indeed, we can provide this data. We did not include them in the first version to avoid raising more questions than answers, as the role of separase in cytokinesis was not the focus of this study. To our (and probably the reviewer's!) surprise, separase S1121A (the mouse equivalent to human S1126A) still promotes polar body extrusion. This indicates that separase promotes polar body extrusion and cytokinesis in meiosis I by a mechanism that is independent of Cyclin B1 binding and inhibition of Cdk1 kinase activity (or protease activity, Kudo et al. 2006). We include this data now in the revised version of the manuscript (Figure EV1B) and mention this result in the main text.

2) Surprisingly, in the discussion the authors compare their results with those of Shindo et al., 2022. The model put forward in that paper is the following: Sole inhibition of separase by securin in metaphase » SAC satisfaction and APC/C activation » liberation of separase, which then cleaves itself without touching cohesin » re-inhibition of separase by CCC » more APC/C activity and more cyclin B1 degradation » 2nd liberation of separase which now finally has a taste for cohesin. I think that everyone in the field, except for maybe the authors, would agree that this cannot be true. This model also does not fit what Wassmann and co-workers observe in MI. What they see is: Redundant inhibition of separase by securin and CCC in metaphase of MI » APC/C activation » separation of bivalents into dyads » re-inhibition of separase by residual CCC during the transition into MII. This is, however, exactly what Stemmann and co-workers observed for mitosis (Hellmuth et al., 2014, 2015). In trying to explain the re-formation of separase-CCC, El Jailani et al. then continue to speculate that S1126 phosphorylation is removed by separase-associated PP2A. While this is an obvious model, there is no evidence for it. (In fact, we have tried very hard to prove this but eventually had to apprehend that this is not what is happening.) In contrast, there is published evidence that dephosphorylation of cyclin B1 at the end of mitosis vastly increases the affinity of residual CCC towards separase (Hellmuth et al., 2015).

This seems to be a misunderstanding, we do not exclude Cyclin B1 as a separase inhibitor before metaphase I. We write "*We found that in oocytes, either securin or cyclin B1 is sufficient to maintain separase under control until metaphase I. At the transition from meiosis I into meiosis II, cyclin B1 plays a major role in separase inhibition, suggesting a hand-over from securin-dependent inhibition to cyclin B1, such as proposed in mitosis.*" If loss of cyclin B1-dependent inhibition of separase but not that of securin has a phenotype after meiosis I, this means that cyclin B1 is the main inhibitor at the transition. However, as we do not know from our experiments whether a given molecule of separase previously inhibited by securin is now inhibited by cyclin B1 we refrain in the revised version from using the term "hand-over", also in the introduction.

We will not be able to refer to unpublished results mentioned by this reviewer, nor refute data published by Shindo et al, 2022 in our discussion. As our manuscript does not address how PP2A binding or autocleavage regulate separase activity or binding of Cyclin B1 we decided to remove the corresponding sentences and leave this apparently rather controversial issue to the experts studying separase structure and activity.

Minor points:

A third inhibitor of separase, namely Sgo2-Mad2, is present in somatic cells arrested in mitosis due to SAC activation (Hellmuth et al., 2020). This sentence may make the impression to the reader that separase inhibition by Sgo2-Mad2 in human somatic cells is limited to mitosis. But it is not....

We respectfully disagree on this point with the reviewer. For up to now, there is no indication that Sgo2-Mad2 contributes to separase inhibition in (oocyte) meiosis. On the contrary, a recent study by Wetherall et al. (PloS Biol. 2025) shows that Sgo2-Mad2 is not contributing to separase inhibition in meiosis I. Hence, for now this inhibitory mechanism has only been described in somatic cells.

Page 5: 'It has been proposed that pericentromeric cohesin protection is maintained until metaphase II to prevent separase from cleaving this fraction of cohesin holding sister chromatids together, as separase may become reactivated upon degradation of cyclin B1 and securin at meiosis I exit.'

Can the authors cite the corresponding study?

This paragraph has been reformulated to introduce the SUMO pathway (in response to reviewer 3), and to better distinguish meiosis I-to-meiosis II transition and metaphase II arrest. The corresponding references have been added.

The authors observe that Mad2 staining was reduced in metaphase I even when bivalents were prematurely separated into sisters due to expression of separase-S1126A in separase-/- securin-/- oocytes. However, Mad2 staining could be rescued by nocodazole treatment. Does that mean that the SAC in meiosis I oocytes senses only attachment and no tension?

We and others have shown previously that the SAC is maintained active only transiently in mouse oocytes. Early and late prometaphase I are separated by 3 hours, and this may be too long to keep the SAC active. In meiosis II it has been shown that with time, single sister chromatids can establish merotelic attachments to both poles that are under tension and thus escape error correction and SAC control. We do not know whether the same can occur in meiosis I, e.g. how single sister chromatids form attachments. Most likely, the decrease in Mad2 staining is due to the combination of the SAC getting inactivated over time, and attachments of sisters to both poles, thus evading the SAC (and error correction). The key result of this experiment is the fact that Mad2 can be recruited to single chromatids indicating that cohesin has been cleaved without APC/C activation. Nocodazole treatment is rather unphysiological and here only used as a positive control to show that in principle, Mad2 can still be recruited and that our stainings work. Concerning SAC activation upon missing tension in meiosis I, we kindly refer the reviewer to our previous study (Vallot et al., Current Biology 2018).

Page 10: ' Using sep-/- oocytes, we set out to determine whether cyclin B1 alone is indeed not required for separase inhibition in meiosis II.'

This sentence would be easier to understand without the 'alone'.

As Cdk1 inactivation is required for anaphase movement, I recommend replacing segregation by separation in the subheading 'APC/C activation is not required for sister chromatid segregation in meiosis II'.

Typo on page 13: 'constitutive active'.

The above editing comments have been addressed.

Page 14: 'However, to address the role of Sgo2-Mad2 convincingly, it will be necessary to invalidate Sgo2 or Mad2 specifically in metaphase I and not before, and find ways to separate a potential role in separase inhibition from other essential functions both proteins play in oocyte meiosis, such as cohesin protection, chromosome alignment and SAC control.'

At least for human Sgo2 and separase, such mutants have been identified (Hellmuth et al., 2020).

2 out of 3 sites in separase that are required for Mad2-Sgo2 binding and that have been described in Hellmuth et al. also affect PP2A binding. Mutations in these two sites lead to precocious sister chromatid segregation (PSCS). However, a mutant of the one site that only affects PP2A binding was not analysed for PSCS under the same conditions. Hence, in our opinion more experiments are required to separate the function of PP2A binding and interaction with Mad2-Sgo2, before performing rescue experiments, optimally in Separase-Sgo2 double knock-out oocytes.

Referee #3:

This manuscript investigates the regulation of separase activity during meiosis in oocytes, focusing on two known separase inhibitors, cyclin B1 and securin.

Firstly, the authors used a separase knockout mouse model and expressed a mutant form of separase (S1121A), which is not inhibited by cyclin B1, in oocytes. Although oocytes expressing separase S1121A normally segregated chromosomes during meiosis I, they resulted in precocious separation of sister chromatids by metaphase II. This indicates that cyclin B1-mediated separase inhibition is not necessary for meiosis I but is essential for maintaining centromeric cohesion until metaphase II. Based on this observation, the authors claim that centromeric cohesion protection is absent during the transition period from meiosis I to metaphase II. However, I am concerned about this claim, as I will discuss later (Major comment 1).

Secondly, they used a securin knockout mouse model and showed that oocytes lacking securin exhibited both meiosis I and meiosis II almost normally. These observations demonstrate that securin is not essential for meiosis, which is consistent with previous studies.

Thirdly, the authors tested oocytes lacking both cyclin B1-mediated and securin-mediated separase inhibitions by combining separase and securin knockouts. These oocytes showed precocious chromosome separation from bivalents to dyads or single chromatids during early prometaphase I. Based on this, the authors claim that centromeric cohesion protection is absent during early prometaphase, but I cannot be convinced of the appropriateness of this claim (Major comment 2).

Fourthly, the authors investigated how separase is inhibited during metaphase II. They used an approach where separase S1121A was introduced after culturing separase knockout oocytes to metaphase II. Separase S1121A did not induce chromosome separation, likely because it was inhibited by securin. Consistently, when both separase and securin were knocked out and separase S1121A was introduced at metaphase II, chromosome separation was induced. Although separation from bivalents to dyads was observed, separation to single chromatids did not occur, suggesting that centromeric cohesion protection was extended to metaphase II. The authors claim that this extension is due to the absence of kinetochore individualization during the transition from meiosis I to meiosis II, but I could not find evidence for this (Major comment 3).

Fifthly, the authors introduced separase S1121A in securin knockout oocytes after culturing to metaphase II. These oocytes exhibited precocious separation of dyad chromosomes to single chromatids. Based on this observation, the authors claim that centromeric cohesion is absent during metaphase II. However, once again, I am not convinced of the

appropriateness of this claim (Major comment 4).

Overall, this paper uses a very clean experimental system taking advantage of separase and securin knockout oocytes to clearly demonstrate the regulation of separase inhibition by cyclin B1 and securin throughout meiosis. While neither cyclin B1 nor securin is essential for inhibiting separase on their own, their combined disruption leads to separase activation, a phenotype expected from previous reports. The authors base their conclusions about the presence or absence of centromeric cohesion protection at various stages of meiosis on the patterns of chromosome separation caused by ectopic separase activation, which is reflected in the title and overall claim throughout the paper. However, as discussed below, I have reservations about their conclusions on the absence of centromeric cohesion protection at several stages of meiosis. I appreciate the beauty of this paper as a clear and robust demonstration of separase inhibition by double inhibitors cyclin B1 and securin in mouse oocytes, which is a great contribution to the field. However, I feel that the paper would benefit from removing or toning down their conclusion about the absence of centromeric cohesion protection at some meiotic stages.

We thank this reviewer for his/her insightful comments. Please see below our reply to the points raised. We will reply first to the major comment 3, because our data showing that centromeric cohesin deprotection requires prior kinetochore individualization is key in our response to the other main comments.

General comment:

We agree with this reviewer that prolonged exposure to active separase will lead to cleavage of cohesin with time, protected or unprotected. This is now mentioned in the manuscript, and we also state that "robust" cohesin protection is absent in meiosis II (title and throughout the text), thus toning down the message as suggested by the reviewer.

3- In the experiments shown in Figure 5, the authors demonstrate that when separase activation occurs for the first time at metaphase II, separation from bivalents to dyads occurs, but not to single chromatids. This suggests that centromeric cohesion protection is extended to metaphase II in this experimental condition. The authors claim that this extension is due to the absence of separase-dependent kinetochore individualization during the transition from meiosis I to metaphase II. However, I cannot find evidence to support this. What the actual data show is that passing through meiosis I without separase activation leads to the extension of centromeric cohesion protection to metaphase II. That is, separase activity is needed to cancel centromeric cohesion protection during the meiosis I-II transition. To support their claim that this cancellation is mediated by separase-dependent kinetochore individualization, the authors should show that forced kinetochore individualization in the absence of separase activity results in a cancellation of centromeric cohesion protection.

As kinetochore individualization requires separase activity (such as we have shown previously, Gryanzova et al. *Embo J.* 2021) we cannot generate bivalents in meiosis II with separated sister kinetochores in absence of separase activity, as suggested by this reviewer. Also, for up to now we do not know the identity of the separase substrate holding the two sister kinetochores close together, thus we cannot force kinetochore individualization in absence of separase activity by for example depleting this substrate in separase knock-out meiosis II oocytes (the substrate could be either Meikin, and/or a fraction of Rec8). However, in our previous study we have done the reverse experiment: we have generated bivalents with separated sister kinetochores in meiosis II oocytes. Under these conditions (e.g. when kinetochores of bivalents are individualized), paired chromosomes separate into sister chromatids. Thus, the visual separation of kinetochores in metaphase II correlates with the ability to segregate sisters. Bivalents with fused kinetochores segregate into dyads even in meiosis II and importantly, those dyads show individualized sister kinetochores. I also want to mention that a study by the Nasmyth lab proposed a third fraction of Rec8 at the centromere to be cleaved in anaphase I, concomitant with a visual separation of sister kinetochores, and they show that this centromeric Rec8 cleavage is required to allow cleavage of Rec8 at the pericentromere (Ogushi et al, *Dev Cell* 2021). For better clarity we added images of separase knock-out oocytes upon rescue with wildtype separase in meiosis II (Fig. EV4B), and we also refer to the study by Ogushi et al. In conclusion, there is no doubt that molecular events depending on separase cleavage activity and visible as kinetochore individualization in anaphase I contribute to deprotection of pericentromeric cohesin in the following division.

1- The authors claim that centromeric cohesion protection is absent during the transition from meiosis I to meiosis II, based on their observation that separase S1121A causes precocious chromosome separation by metaphase II. However, I would interpret their observation to mean that centromeric cohesion protection does exist but centromeric cohesion is gradually lost due to prolonged separase activity, leading to precocious chromosome separation. The difference in interpretation between the authors and myself might stem from a difference in assumptions about centromeric cohesion protection. The authors might assume that if centromeric cohesion is protected, there should be no reduction in cohesin at the centromeres at all. However, this assumption is less likely to be correct. A more general assumption is that if centromeric cohesion is protected, there is less reduction in cohesin at centromeres compared to chromosome arms. If this more likely assumption is considered, it is natural that centromeric cohesion would eventually be lost if separase activity is prolonged by S1121A even in the presence of centromeric cohesion protection.

We agree that an alternative interpretation of our results could be that too much of separase activity at the transition from meiosis I-to-meiosis II may eventually lead to weakening of cohesion with time. Also, in the light of the SUMO pathway contributing to

cohesin maintenance, gradual decrease of cohesin at the centromere with time may lead to the observed phenotype. Thus, we refrain in the revised version from stating that cohesin protection is absent, but instead we state that proper maintenance of cohesin requires inhibition of separase by cyclin B1.

2- The authors claim that centromeric cohesion protection is absent during early prometaphase I, based on their observation that chromosome bivalents are precociously separated into dyads or single chromatids during early prometaphase I when both cyclin B1-mediated and securin-mediated inhibitions are simultaneously disrupted. For the same reasons mentioned above, I cannot be convinced that this claim is appropriate. Indeed, their data show that there is a fraction of chromosomes that resulted in dyads, consistent with the idea that chromosomes separated from bivalents to dyads, and subsequently from dyads to single chromatids. This sequential chromosome separation would support the presence of centromeric cohesion protection.

The fact that Sgo2 and PP2A are largely absent from the centromere region at meiosis resumption, and the speed by which chromosomes and sister chromatids fly apart leaves no doubt in our opinion that centromeric cohesin protection is absent. However, we agree that arm cohesin is removed before centromeric cohesin, indicating that the sequence of events stays the same. But we also know that centromeric cohesin is cleaved in anaphase I after arm cohesin, even though it is pericentromeric and not centromeric cohesin that is protected. Thus, in our opinion, cohesin protection is more than just the sequence of events, pericentromeric cohesin cleavage is actively prevented by Sgo2-PP2A dependent cohesin protection and requires kinetochore individualization to be abrogated. (please see also reply to major comment 4)

4- In the experiments shown in Figure 6, the authors show that a second round separase activation at metaphase II results in precocious separation of chromosome dyads into single chromatids. Based on this observation, the authors claim that centromeric cohesion is absent during metaphase II. However, for the same reasons mentioned in comment 1, I am afraid that this claim is not appropriate. If ectopic separase activity is present, it is natural that even if centromeric cohesion protection exists, it would eventually lead to the separation into single chromatids.

If the prediction of this reviewer were correct, chromosomes with protected pericentromeric cohesin would segregate into dyads, and upon further presence of active separase, into sisters. We propose that without kinetochore individualization, pericentromeric cohesin remains protected, and bivalents would only segregate into dyads, even after prolonged incubation with constitutively active separase. Using the experimental setting described in Figure 5, we were able to test which prediction was correct:

sec^{-/-}sep^{-/-} double KO oocytes in metaphase II were injected with separase S1121A. We compared the number of dyads and sister chromatids 4 hours and 8 hours post injection. In both cases we found the same number of dyads, and hardly any sister chromatids. Thus, also with time we did not observe sister chromatid separation, which would indicate pericentromeric cohesin cleavage. Therefore, we estimate that our experimental setting allows us to conclude whether cohesin protection is present or absent. However, on suggestion of this reviewer, we also take into account that prolonged or unphysiological expression of separase may lead to overall cohesin weakening (such as described in Chiang et al. 2011, Biol of Reprod.) and thus, weakening of pericentromeric cohesin. Thus, in the revised version of the manuscript we now state that "robust" cohesin protection is absent in metaphase II.

Concerning absence of protection at resumption of meiosis I, our experiment in metaphase II shows that cohesin protection cannot be overridden after at least 8 hour exposure to active separase, largely exceeding the time window when sister chromatid segregation took place in early prometaphase I (< 3hours).

5- One of the main targets of this study is mechanisms of centromeric cohesion protection during the transition from meiosis I to meiosis II in mouse oocytes. One such mechanism known during this period is the centromeric SUMO pathway (PMID: 29754905), yet the authors ignore this paper throughout their manuscript.

We apologize for not having taken into consideration this mechanism, and thank the reviewer for point this out. We both introduce and discuss the SUMO pathway in light of our results in the revised version of the manuscript.

Minor comment:

It would be beneficial to add quantification of the separase activity sensor in Fig. 3A to strengthen the results.

We refrained from quantifying separase activity, because unlike in the other conditions, onset occurred so early in the separase S1121A injected oocytes that we couldn't properly set a value to start with. Additionally, chromosomes are not properly condensed yet at this stage, making it impossible to correctly quantify the sensor and compare it to the control.

Dr. Katja Wassmann
CNRS - Institut Jacques Monod
Mechanisms of Meiosis
15 rue Helene Brion
PARIS CEDEX 13, Paris 75013
France

14th Jul 2025

Re: EMBOJ-2025-120451R
Eliminating separase inhibition reveals absence of robust cohesin protection in oocyte metaphase II

Dear Katja,

Thank you for submitting your revised manuscript to The EMBO Journal. Two of the original referees have now assessed it once more, and were generally satisfied with the revisions. Referee 2 retains a few presentational concerns that would need to be incorporated during a final round of minor revision. In addition, please also address the following remaining editorial issues at this stage

- Please carefully go through the reference list and make sure that each reference is complete with citation year, volume, and page/locator numbers - this information is currently missing for several of them.
- Please rename the Material & Methods section simply into "Methods"
- Please rename the Conflict of Interest section into "Disclosure and Competing Interests Statement", in accordance with our updated Guide to Authors (<https://www.embopress.org/competing-interests>)
- As we are switching from a free-text author contribution statement towards a more formal statement based on Contributor Role Taxonomy (CRediT) terms, please remove the present Author Contribution section and instead specify each author's contribution(s) directly in the Author Information page of our submission system during upload of the final manuscript. See <https://casrai.org/credit/> for more information.
- On the title page of the Appendix, please include the header "Appendix for [manuscript title]"

I am returning the manuscript to you for a final round of minor revision, solely to allow you to make these modifications and upload the revised files. Once we will have received them, we should be ready to swiftly proceed with formal acceptance and production of the manuscript.

With kind regards,

Hartmut

- 1) Every manuscript requires a Data Availability section (even if only stating that no deposited datasets are included). Primary datasets or computer code produced in the current study have to be deposited in appropriate public repositories prior to resubmission, and reviewer access details provided in case that public access is not yet allowed. Further information: embopress.org/page/journal/14602075/authorguide#dataavailability
- 2) Each figure legend must specify
 - size of the scale bars that are mandatory for all micrograph panels
 - the statistical test used to generate error bars and P-values
 - the type error bars (e.g., S.E.M., S.D.)

- the number (n) and nature (biological or technical replicate) of independent experiments underlying each data point
- Figures may not include error bars for experiments with $n < 3$; scatter plots showing individual data points should be used instead.

9) To facilitate reproducibility and cross-laboratory adoption of methodologies, please structure the Materials & Methods section as outlined in our guide to authors, including a completed Reagents and Tools Table that can be downloaded from our author guidelines as well (<https://www.embopress.org/page/journal/14602075/authorguide#structuredmethods>).

10) Digital image enhancement is acceptable practice, as long as it accurately represents the original data and conforms to community standards. If a figure has been subjected to significant electronic manipulation, this must be clearly noted in the figure legend and/or the 'Materials and Methods' section. The editors reserve the right to request original versions of figures and the original images that were used to assemble the figure. Finally, we generally encourage uploading of numerical as well as gel/blot image source data; for details see: embopress.org/page/journal/14602075/authorguide#sourcedata

In the interest of ensuring the conceptual advance provided by the work, we recommend submitting a revision within 3 months (12th Oct 2025). Please discuss the revision progress ahead of this time with the editor if you require more time to complete the revisions. Use the link below to submit your revision:

Link Not Available

Referee #2:

I was enthusiastic about this study already in its first version. In the revised version, Wassmann and coworkers adequately addressed my major point. As outlined below, a few of my other points have not been addressed as I deem necessary but these were merely commentaries and did not involve additional experiments. I suggest that the authors check once more whether they always cite the fitting literature and whether their statements always accurately reflect the cited work. After that, the study at hand is truly fit for publication in the EMBO Journal.

i) My previous commentary:

Surprisingly, in the discussion the authors compare their results with those of Shindo et al., 2022. The model put forward in that paper is the following: Sole inhibition of separase by

securin in metaphase » SAC satisfaction and APC/C activation » liberation of separase, which then cleaves itself without touching cohesin » re-inhibition of separase by CCC » more APC/C activity and more cyclin B1 degradation » 2nd liberation of separase which now finally has a taste for cohesin. I think that everyone in the field, except for maybe the authors, would agree that this cannot be true. This model also does not fit what Wassmann and co-workers observe in MI. What they see is: Redundant inhibition of separase by securin and CCC in metaphase of MI » APC/C activation » separation of bivalents into dyads » reinhibition of separase by residual CCC during the transition into MII. This is, however, exactly what Stemmann and co-workers observed for mitosis (Hellmuth et al., 2014, 2015). In trying to explain the re-formation of separase-CCC, El Jailani et al. then continue to speculate that S1126 phosphorylation is removed by separase-associated PP2A. While this is an obvious model, there is no evidence for it. (In fact, we have tried very hard to prove this but eventually had to apprehend that this is not what is happening.) In contrast, there is published evidence that dephosphorylation of cyclin B1 at the end of mitosis vastly increases the affinity of residual CCC towards separase (Hellmuth et al., 2015).

The authors' reply:

This seems to be a misunderstanding, we do not exclude Cyclin B1 as a separase inhibitor before metaphase I. We write "We found that in oocytes, either securin or cyclin B1 is sufficient to maintain separase under control until metaphase I. At the transition from meiosis I into meiosis II, cyclin B1 plays a major role in separase inhibition, suggesting a hand-over from securin-dependent inhibition to cyclin B1, such as proposed in mitosis." If loss of cyclin B1-dependent inhibition of separase but not that of securin has a phenotype after meiosis I, this means that cyclin B1 is the main inhibitor at the transition. However, as we do not know from our experiments whether a given molecule of separase previously inhibited by securin is now inhibited by cyclin B1 we refrain in the revised version from using the term "handover", also in the introduction.

We will not be able to refer to unpublished results mentioned by this reviewer, nor refute data published by Shindo et al, 2022 in our discussion. As our manuscript does not address how PP2A binding or autocleavage regulate separase activity or binding of Cyclin B1 we decided to remove the corresponding sentences and leave this apparently rather controversial issue to the experts studying separase structure and activity.

My answer to the authors' reply:

A misunderstanding indeed. I did not suggest to refer to unpublished results nor did I talk about regulation of separase by PP2A or autocleavage.

What I tried to explain was that Shindo et al., 2022, propose a switch from securin- to CCC-dependent inhibition of separase PRIOR to onset of anaphase and that this is different from what Wassmann and co-workers observe in murine female meiosis. In contrast, their observation reflect much more what was reported by Hellmuth et al.. In their 2015 Mol Cell paper, these authors show that separase-securin and separase-CCC complexes coexist in prometaphase. From these, separase is liberated and activated before separase-CCC complexes form anew in late mitosis. (They continue to demonstrate that formation of this 2nd peak of separase-CCC at a time when cyclin B1 levels are already very low is facilitated by dephosphorylation of cyclin B1, which could also explain the re-inhibition of separase after female meiosis I.)

In their revised MS, Wassmann and coworkers now write: "At the transition from meiosis I into meiosis II, cyclin B1 plays a major role in separase inhibition, suggesting that securin-dependent inhibition switches to cyclin B1 AFTER metaphase-to-anaphase transition, such as proposed in mitosis (Yu et al., 2023)." Once again: This does not accurately cite the work of Shindo and co-workers, who propose that this switch occurs BEFORE the activation of separase!

ii) My previous commentary:

A third inhibitor of separase, namely Sgo2-Mad2, is present in somatic cells arrested in mitosis due to SAC activation (Hellmuth et al., 2020). This sentence may make the impression to the reader that separase inhibition by Sgo2-Mad2 in human somatic cells is limited to mitosis. But it is not....

The authors' reply:

We respectfully disagree on this point with the reviewer. For up to now, there is no indication that Sgo2-Mad2 contributes to separase inhibition in (oocyte) meiosis. On the contrary, a recent study by Wetherall et al. (PloS Biol. 2025) shows that Sgo2-Mad2 is not contributing to separase inhibition in meiosis I. Hence, for now this inhibitory mechanism has only been described in somatic cells.

My answer to the authors' reply:

Another misunderstanding. I was not referring to meiosis and apologize for not having made this clearer. What I meant is that

the separase-Sgo2-Mad2 complex in human somatic cells is not limited to mitosis but exists throughout the cell cycle (except for late M/early G1), which is consistent with Rodriguez-Bravo et al., 2014 who find that Mad2 is activated at NPCs during interphase.

iii) Page 14: 'However, to address the role of Sgo2-Mad2 convincingly, it will be necessary to invalidate Sgo2 or Mad2 specifically in metaphase I and not before, and find ways to separate a potential role in separase inhibition from other essential functions both proteins play in oocyte meiosis, such as cohesin protection, chromosome alignment and SAC control.'

At least for human Sgo2 and separase, such mutants have been identified (Hellmuth et al., 2020).

The authors' reply:

2 out of 3 sites in separase that are required for Mad2-Sgo2 binding and that have been described in Hellmuth et al. also affect PP2A binding. Mutations in these two sites lead to precocious sister chromatid segregation (PSCS). However, a mutant of the one site that only affects PP2A binding was not analysed for PSCS under the same conditions. Hence, in our opinion more experiments are required to separate the function of PP2A binding and interaction with Mad2-Sgo2, before performing rescue experiments, optimally in Separase-Sgo2 double knock-out oocytes.

My answer to the authors' reply:

I agree that these additional experiments would be beyond the scope of the study at hand (but just to clarify: Hellmuth et al. identified only 2 (not 3) separase variants that are defective in Sgo2-Mad2 binding.

Referee #3:

The authors have carefully considered the points I raised and made appropriate revisions to the wording. In particular, they have toned down their wording regarding the presence or absence of cohesin protection, which has made the claims of the manuscript more accurate and valuable. While I am not fully convinced about the authors' claim that kinetochore individualization contributes to cohesin deprotection, this claim is based on the conclusions of their previously published work. Therefore, I see no issue with the current manuscript including arguments that are built upon this premise.

Rebuttal letter

Referee #2:

I was enthusiastic about this study already in its first version. In the revised version, Wassmann and coworkers adequately addressed my major point. As outlined below, a few of my other points have not been addressed as I deem necessary but these were merely commentaries and did not involve additional experiments. I suggest that the authors check once more whether they always cite the fitting literature and whether their statements always accurately reflect the cited work. After that, the study at hand is truly fit for publication in the EMBO Journal.

i) My previous commentary:

Surprisingly, in the discussion the authors compare their results with those of Shindo et al., 2022. The model put forward in that paper is the following: Sole inhibition of separase by securin in metaphase » SAC satisfaction and APC/C activation » liberation of separase, which then cleaves itself without touching cohesin » re-inhibition of separase by CCC » more APC/C activity and more cyclin B1 degradation » 2nd liberation of separase which now finally has a taste for cohesin. I think that everyone in the field, except for maybe the authors, would agree that this cannot be true. This model also does not fit what Wassmann and co-workers observe in MI. What they see is: Redundant inhibition of separase by securin and CCC in metaphase of MI » APC/C activation » separation of bivalents into dyads » reinhibition of separase by residual CCC during the transition into MII. This is, however, exactly what Stemmann and co-workers observed for mitosis (Hellmuth et al., 2014, 2015).

In trying to explain the re-formation of separase-CCC, El Jailani et al. then continue to speculate that S1126 phosphorylation is removed by separase-associated PP2A. While this is an obvious model, there is no evidence for it. (In fact, we have tried very hard to prove this but eventually had to apprehend that this is not what is happening.) In contrast, there is published evidence that dephosphorylation of cyclin B1 at the end of mitosis vastly increases the affinity of residual CCC towards separase (Hellmuth et al., 2015).

The authors' reply:

This seems to be a misunderstanding, we do not exclude Cyclin B1 as a separase inhibitor before metaphase I. We write "We found that in oocytes, either securin or cyclin B1 is sufficient to maintain separase under control until metaphase I. At the transition from meiosis I into meiosis II, cyclin B1 plays a major role in separase inhibition, suggesting a hand-over from securin-dependent inhibition to cyclin B1, such as proposed in mitosis." If loss of cyclin B1-dependent inhibition of separase but not that of securin has a phenotype after meiosis I, this means that cyclin B1 is the main inhibitor at the transition. However, as we do not know from our experiments whether a given molecule of separase previously inhibited by securin is now inhibited by cyclin B1 we refrain in the revised version from using the term "handover", also in the introduction.

We will not be able to refer to unpublished results mentioned by this reviewer, nor refute data published by Shindo et al, 2022 in our discussion. As our manuscript does not address how PP2A binding or autocleavage regulate separase activity or binding of Cyclin B1 we decided to remove the corresponding sentences and leave

this apparently rather controversial issue to the experts studying separase structure and activity.

My answer to the authors' reply:

A misunderstanding indeed. I did not suggest to refer to unpublished results nor did I talk about regulation of separase by PP2A or autocleavage.

What I tried to explain was that Shindo et al., 2022, propose a switch from securin- to CCC-dependent inhibition of separase PRIOR to onset of anaphase and that this is different from what Wassmann and co-workers observe in murine female meiosis. In contrast, their observation reflect much more what was reported by Hellmuth et al.. In their 2015 Mol Cell paper, these authors show that separase-securin and separase-CCC complexes coexist in prometaphase. From these, separase is liberated and activated before separase-CCC complexes form anew in late mitosis. (They continue to demonstrate that formation of this 2nd peak of separase-CCC at a time when cyclin B1 levels are already very low is facilitated by dephosphorylation of cyclin B1, which could also explain the re-inhibition of separase after female meiosis I.) In their revised MS, Wassmann and coworkers now write: " to cyclin B1 AFTER metaphase-to-anaphase transition, such as proposed in mitosis (Yu et al., 2023)." Once again: This does not accurately cite the work of Shindo and co-workers, who propose that this switch occurs BEFORE the activation of separase!

We have not adressed whether there is a switch or if that switch occurs before or after Separase activation (at anaphase onset), and it is quite possible that there are distinct fractions of Separase regulated differently (and our preliminary data indicate that this might indeed be the case). From our data in the present manuscript we also cannot conclude that securin and cyclin B1 inhibited separase co-exists in vivo, we can only conclude that the two inhibitory mechanisms are redundant, and one can take over in absence of the other. I will refrain from the overinterpretation of our data or taking sides on this issue, and this was also not the focus of this study. The sentence has thus been reformulated to simplify and avoid any further misunderstanding:

"At the transition from meiosis I into meiosis II, cyclin B1 plays a major role in separase inhibition, suggesting that securin- dependent inhibition is not important at this meiotic cell cycle stage."

We are not citing Yu et al 2023 in this specific context anymore, which -by the way- is only a review summarizing the literature on separase inhibition.

ii) My previous commentary:

A third inhibitor of separase, namely Sgo2-Mad2, is present in somatic cells arrested in mitosis due to SAC activation (Hellmuth et al., 2020). This sentence may make the impression to the reader that separase inhibition by Sgo2-Mad2 in human somatic cells is limited to mitosis. But it is not....

The authors' reply:

We respectfully disagree on this point with the reviewer. For up to now, there is no indication that Sgo2-Mad2 contributes to separase inhibition in (oocyte) meiosis. On the contrary, a recent study by Wetherall et al. (PloS Biol. 2025) shows that Sgo2-Mad2 is not contributing to separase inhibition in meiosis I. Hence, for now this inhibitory mechanism has only been described in somatic cells.

My answer to the authors' reply:

Another misunderstanding. I was not referring to meiosis and apologize for not having made this clearer. What I meant is that the separase-Sgo2-Mad2 complex in human somatic cells is not limited to mitosis but exists throughout the cell cycle (except for late M/early G1), which is consistent with Rodriguez-Bravo et al., 2014 who find that Mad2 is activated at NPCs during interphase.

The paper by Rodriguez-Bravo et al., 2014 indeed shows that during interphase, Mad1 directs the assembly of MCC complexes for a premitotic "Wait-Anaphase" signal. However, it is currently unknown whether this is also related to potential formation of separase-Sgo2-Mad2 complexes prior to mitosis. To my best of knowledge, the existence of ternary separase-Sgo2-Mad2 complexes outside mitosis has not been demonstrated yet.

iii) Page 14: 'However, to address the role of Sgo2-Mad2 convincingly, it will be necessary to invalidate Sgo2 or Mad2 specifically in metaphase I and not before, and find ways to separate a potential role in separase inhibition from other essential functions both proteins play in oocyte meiosis, such as cohesin protection, chromosome alignment and SAC control.'

At least for human Sgo2 and separase, such mutants have been identified (Hellmuth et al., 2020).

The authors' reply:

2 out of 3 sites in separase that are required for Mad2-Sgo2 binding and that have been described in Hellmuth et al. also affect PP2A binding. Mutations in these two sites lead to precocious sister chromatid segregation (PSCS). However, a mutant of the one site that only affects PP2A binding was not analysed for PSCS under the same conditions. Hence, in our opinion more experiments are required to separate the function of PP2A binding and interaction with Mad2-Sgo2, before performing rescue experiments, optimally in Separase-Sgo2 double knock-out oocytes.

My answer to the authors' reply:

I agree that these additional experiments would be beyond the scope of the study at hand (but just to clarify: Hellmuth et al. identified only 2 (not 3) separase variants that are defective in Sgo2-Mad2 binding.

Indeed, there are only two separase variants identified. However, this is not discussed in the manuscript, therefore no changes were necessary.

Referee #3:

The authors have carefully considered the points I raised and made appropriate revisions to the wording. In particular, they have toned down their wording regarding the presence or absence of cohesin protection, which has made the claims of the manuscript more accurate and valuable. While I am not fully convinced about the authors' claim that kinetochore individualization contributes to cohesin deprotection, this claim is based on the conclusions of their previously published work. Therefore, I

see no issue with the current manuscript including arguments that are built upon this premise.

I thank the reviewer for acknowledging that we have addressed his/her concerns. I hope our future work will further convince this reviewer that kinetochore individualization through cleavage of a separate substrate after metaphase to anaphase transition in meiosis I is a key event for deprotection of centromeric cohesin in meiosis II.

Dr. Katja Wassmann
CNRS - Institut Jacques Monod
Mechanisms of Meiosis
15 rue Helene Brion
PARIS CEDEX 13, Paris 75013
France

18th Jul 2025

Re: EMBOJ-2025-120451R1
Eliminating separase inhibition reveals absence of robust cohesin protection in oocyte metaphase II

Dear Katja,

Thank you for submitting your final revised manuscript for our consideration. I am pleased to inform you that we have now accepted it for publication in The EMBO Journal.

With kind regards,

Hartmut
